# Understanding $\ell^4$-based Dictionary Learning: Interpretation, Stability, and Robustness

**Yuexiang Zhai**[1,2*]  **Hermish Mehta**[1]  **Zhengyuan Zhou**[3]  **Yi Ma**[1]
[1]Department of EECS, UC Berkeley    [2]ByteDance Inc.    [3]Stern School of Business, NYU

## Abstract

Recently, the $\ell^4$-norm maximization has been proposed to solve the sparse dictionary learning (SDL) problem. The simple MSP (matching, stretching, and projection) algorithm proposed by Zhai et al. (2019a) has shown to be surprisingly efficient and effective. This paper aims to better understand this algorithm from its strong geometric and statistical connections with the classic PCA and ICA, as well as their associated fixed-point style algorithms. Such connections provide a unified way of viewing problems that pursue *principal*, *independent*, or *sparse* components of high-dimensional data. Our studies reveal additional good properties of $\ell^4$-maximization: not only is the MSP algorithm for sparse coding insensitive to small noise, but it is also robust to outliers and resilient to sparse corruptions. We provide statistical justification for such inherently nice properties. To corroborate the theoretical analysis, we also provide extensive and compelling experimental evidence with both synthetic data and real images.

## 1 Introduction

The explosion of massive amounts of high-dimensional data has become the modern-day norm for a large number of scientific and engineering disciplines and hence presents a daunting challenge for both computation and learning. Rising to this challenge, *sparse dictionary learning* (SDL) provides a potent framework in representation learning that exploits the blessing of dimensionality: real data tends to lie in or near some low-dimensional subspaces or manifolds, even though the ambient dimension is often extremely large (e.g. the number of raw pixels in an image). More specifically, SDL (Olshausen & Field (1997); Mairal et al. (2008; 2012; 2014); Spielman et al. (2012); Sun et al. (2015); Bai et al. (2018); Qu et al. (2019)) concerns the problem of learning a compact, sparse representation from raw, unlabelled data: given a data matrix $\boldsymbol{Y} = [\boldsymbol{y}_1, \boldsymbol{y}_2, \ldots, \boldsymbol{y}_p] \in \mathbb{R}^{n \times p}$ that contains $p$ $n$-dimensional samples, one aims to find a linear transformation (i.e. *a dictionary*) $\boldsymbol{D} \in \mathbb{R}^{n \times m}$ and an associated maximally sparse representation $\boldsymbol{X} = [\boldsymbol{x}_1, \boldsymbol{x}_2, \ldots, \boldsymbol{x}_p] \in \mathbb{R}^{m \times p}$ that satisfies

$$\boldsymbol{Y} = \boldsymbol{D}\boldsymbol{X}. \tag{1}$$

As the data matrix $\boldsymbol{Y}$ can represent a variety of signals (e.g. images, audios, languages, and genetics etc) in practical applications, SDL provides a versatile structure-seeking formulation that has found widespread applications in computational neuroscience, image processing, computer vision, and machine learning at large (Olshausen & Field, 1996; 1997; Argyriou et al., 2008; Ranzato et al., 2007; Elad & Aharon, 2006; Wright et al., 2008; Yang et al., 2010; Zhang et al., 2013; Mairal et al., 2014; Zhang et al., 2014; 2019).

**Related Works.** Motivated by this practical significance, there has been a growing surge of interest recently (e.g. Rambhatla et al. (2019); Bai et al. (2018); Gilboa et al. (2018); Nguyen et al. (2018); Chatterji & Bartlett (2017); Mensch et al. (2016)) that aims to tackle SDL. In attempts to recover the sparse signals $\boldsymbol{X}$, these existing work adopt an $\ell^0$- or $\ell^1$-penalty function to promote the underlying sparsity and give various optimization algorithms for the resulting objectives (some of those are heuristics while a few others have theoretical convergence guarantees). Although these penalty

---

*Work done when interning at ByteDance. YZ would like to thank Chong You at UC Berkeley for meaningful discussions. Correspondence to: Yuexiang Zhai, ysz@berkeley.edu or Yi Ma, yima@eecs.berkeley.edu. Codes are available at `https://github.com/hermish/ZMZM-ICLR-2020`.

functions are indeed sparsity-promoting, the resulting optimization problems must be solved one row at a time, hence resulting as many optimization problems as the ambient dimension $n$. Consequently, $\ell^0$- or $\ell^1$-based objectives result only in local methods (i.e. cannot yield the entire solution at once) and hence entail prohibitive computational burden. Another prominent approach in SDL is Sum-of-Squares (SOS), proposed by and articulated in a series of recent work (Barak et al. (2015); Ma et al. (2016); Schramm & Steurer (2017)). The key idea there is to utilize the properties of higher order SOS polynomials to correctly recover one column of the dictionary at a time, for which there are $m$ columns in total. However, the computational complexity of these recovery methods are quasi-polynomial, thus again resulting in large computational expense.

Very recently, in the *complete dictionary learning*[1] setting, a novel global approach has been suggested in Zhai et al. (2019a;b) that presents a formulation which can efficiently recover the sparse signal matrix $\boldsymbol{X}$ once and for all. In particular, Zhai et al. (2019b) has shown that if the generative model for $\boldsymbol{Y} = \boldsymbol{D}_o \boldsymbol{X}_o \in \mathbb{R}^{n \times p}$ satisfies that $\boldsymbol{D}_o \in \mathsf{O}(n; \mathbb{R})$ is orthonormal and $\boldsymbol{X}_o \in \mathbb{R}^{n \times p}$ is Bernoulli-Gaussian sparse,[2] then maximizing the $\ell^4$-norm[3] of $\boldsymbol{AY}$ over $\mathsf{O}(n; \mathbb{R})$:

$$\max_{\boldsymbol{A}} \frac{1}{4} \|\boldsymbol{AY}\|_4^4 \quad \text{subject to } \boldsymbol{A} \in \mathsf{O}(n; \mathbb{R}) \quad (\text{or } \boldsymbol{AA}^* = \boldsymbol{I}), \tag{2}$$

is able to find the ground truth dictionary $\boldsymbol{D}_o$ up to an arbitrary signed permutation. Moreover, Zhai et al. (2019b) has proposed the simple "*Matching, Stretching, and Projection*" (MSP) algorithm, which is shown to be experimentally efficient and effective, for solving the program in equation 2:

$$\text{MSP:} \quad \boldsymbol{A}_{t+1} = \mathcal{P}_{\mathsf{O}(n; \mathbb{R})} \left[ (\boldsymbol{A}_t \boldsymbol{Y})^{\circ 3} \boldsymbol{Y}^* \right] = \boldsymbol{U}_t \boldsymbol{V}_t^*, \tag{3}$$

where $\boldsymbol{U}_t \boldsymbol{V}_t^*$ are from the singular value decomposition: $\boldsymbol{U}_t \boldsymbol{\Sigma}_t \boldsymbol{V}_t^* = \text{SVD}[(\boldsymbol{A}_t \boldsymbol{Y})^{\circ 3} \boldsymbol{Y}^*]$.

In this paper, we here give an alternative (arguably simpler and more revealing) derivation of the MSP algorithm (3). Consider the *Lagrangian* formulation of the constrained optimization problem given in equation 2. The necessary condition of critical points, $\nabla_{\boldsymbol{A}} \frac{1}{4} \|\boldsymbol{AY}\|_4^4 = \nabla_{\boldsymbol{A}} \langle \boldsymbol{\Lambda}, \boldsymbol{AA}^* - \boldsymbol{I} \rangle$ for some Lagrangian multipliers $\boldsymbol{\Lambda}$, implies:

$$(\boldsymbol{AY})^{\circ 3} \boldsymbol{Y}^* = (\boldsymbol{\Lambda} + \boldsymbol{\Lambda}^*) \boldsymbol{A}. \tag{4}$$

As the optimization is over the orthogonal group $\mathsf{O}(n; \mathbb{R})$, restricting the condition in equation 4 onto the orthogonal group yields a necessary condition for any critical point $\boldsymbol{A}$:[4]

$$\mathcal{P}_{\mathsf{O}(n; \mathbb{R})} \left[ (\boldsymbol{AY})^{\circ 3} \boldsymbol{Y}^* \right] = \boldsymbol{A}. \tag{5}$$

Hence the critical point $\boldsymbol{A}$ can be viewed as a "*fixed point*" of the map: $\mathcal{P}_{\mathsf{O}(n; \mathbb{R})} \left[ ((\cdot) \boldsymbol{Y})^{\circ 3} \boldsymbol{Y}^* \right]$ from $\mathsf{O}(n; \mathbb{R})$ to itself. The MSP algorithm in equation 3 is to find the fixed point(s) of this map.

Notice that the orthonormal constraint $\boldsymbol{A} \in \mathsf{O}(n; \mathbb{R})$ in equation 2 can be viewed as enforcing the orthogonality of $n$ unit vectors simultaneously. So, more flexibly and generally, one may choose to compute any $k$, for $1 \le k \le n$, leading orthonormal bases of $\boldsymbol{D}_o$ by solving the program:

$$\max_{\boldsymbol{W}} \frac{1}{4} \|\boldsymbol{W}^* \boldsymbol{Y}\|_4^4 \quad \text{subject to } \boldsymbol{W} \in \mathsf{St}(n, k; \mathbb{R}) \subset \mathbb{R}^{n \times k}, \tag{6}$$

where $\mathsf{St}(n, k; \mathbb{R})$ is the *Stiefel manifold*.[5] The orthogonal group $\mathsf{O}(n; \mathbb{R})$ and the unit sphere $\mathbb{S}^{n-1}$ can be viewed as two special cases of the Stiefel manifold $\mathsf{St}(n, k; \mathbb{R})$, with $k = n$ and $k = 1$, respectively. In some specific tasks such as dictionary learning and blind deconvolution, optimization over the unit sphere has been widely practiced, such as in Sun et al. (2015); Bai et al. (2018); Zhang et al. (2018); Kuo et al. (2019). The more general setting of maximizing a convex function over any compact set also has been studied by Journée et al. (2010) in the context of sparse PCA, which has provided convergence guarantees for this class of programs.

---

[1] Complete dictionary learning requires the learned dictionary $\boldsymbol{D}$ in equation 1 to be square and invertible.

[2] Each entry $x_{i,j}$ of $\boldsymbol{X}$ can be represented as the product of a Bernoulli variable and a normal Gaussian variable: $x_{i,j} = \Omega_{i,j} V_{i,j}$, where $\Omega_{i,j} \sim_{iid} \text{Ber}(\theta)$ and $V_{i,j} \sim_{iid} \mathcal{N}(0, 1)$, similar for vectors or scalars. This is the standard setting adopted in Spielman et al. (2012); Sun et al. (2015); Bai et al. (2018).

[3] We abuse the notation a bit, by denoting $\|\cdot\|_4^4$ as the sum of element-wise $4^{\text{th}}$ power of all entries of a vector and matrix, that is, $\forall \boldsymbol{a} \in \mathbb{R}^n, \|\boldsymbol{a}\|_4^4 = \sum_{i=1}^n a_i^4$ and $\forall \boldsymbol{A} \in \mathbb{R}^{n \times m}, \|\boldsymbol{A}\|_4^4 = \sum_{i,j} a_{i,j}^4$.

[4] For any symmetric matrix $\boldsymbol{S} \in \mathbb{R}^{n \times n}$ and an orthogonal matrix $\boldsymbol{A} \in \mathsf{O}(n; \mathbb{R})$, the projection of $\boldsymbol{SA}$ onto the orthogonal group is $\boldsymbol{A}$: $\mathcal{P}_{\mathsf{O}(n; \mathbb{R})}[\boldsymbol{SA}] = \boldsymbol{A}$, one may see Absil & Malick (2012) for details.

[5] For any $1 \le k \le n, \mathsf{St}(n, k; \mathbb{R}) \doteq \{\boldsymbol{W} \in \mathbb{R}^{n \times k} : \boldsymbol{W}^* \boldsymbol{W} = \boldsymbol{I}_k\}$.

**Our Contributions.**   Our contributions are twofold. First, by taking a suitable analytical angle, we reveal novel geometric and statistical connections between PCA, ICA and the $\ell^4$-norm maximization based SDL. We then show that algorithm-wise, the fixed-point type MSP algorithm for $\ell^4$-norm maximization has the same nature as the classic *power-iteration method* for PCA (Jolliffe, 2011) and the FastICA algorithm for ICA (Hyvärinen & Oja, 1997). This interpretation gives a unified view for problems that pursue *principal*, *independent*, or *sparse* components from high-dimensional data and enriches our understanding of low-dimensional structure recovery frameworks, classical and new, at both formulation and algorithmic fronts.

Second, and more importantly from a practical perspective, we examine how MSP performs under a variety of more realistic conditions, when the measurements $\boldsymbol{Y}$ could be contaminated with noise, outliers, or sparse corruptions. We show that, similar to PCA, $\ell^4$-norm maximization and the MSP algorithm are inherently stable to small noise. Somewhat surprisingly though, unlike PCA, the MSP algorithm is further robust to outliers and resilient to sparse gross errors! We provide character-izations of these desirable properties of MSP. The claims are further corroborated with extensive experiments on both synthetic data and real images. Taken as a whole, our results contribute to the broad landscape of dictionary learning by affirming that $\ell^4$-maximization based SDL and the corresponding global algorithm MSP provide a valuable toolkit to the existing literature.

## 2 SDL VERSUS PCA AND ICA

### 2.1 PURSUIT OF PRINCIPAL, INDEPENDENT, OR SPARSE COMPONENTS

**Relation with the Geometric Interpretation of PCA.**   For a data matrix $\boldsymbol{Y} \in \mathbb{R}^{n \times p}$, *Principal Component Analysis* (PCA), aiming to find the top (top $k$) left singular vector (vectors) of $\boldsymbol{Y}$,

$$\max_{\boldsymbol{W}} \frac{1}{2} \left\| \boldsymbol{W}^* \boldsymbol{Y} \right\|_F^2 \quad \text{subject to } \boldsymbol{W} \in \mathsf{St}(n, k; \mathbb{R}), \tag{7}$$

can be understood as finding a direction (a $k$-dimensional subspace) in $\mathrm{row}(\boldsymbol{Y})$ in which $\boldsymbol{Y}$ has the largest $\ell^2$-norm (Frobenius norm). For instance, finding the direction with the largest $\ell^2$-norm over the unit sphere can be viewed as calculating the spectral norm (or the largest singular value), of matrix $\boldsymbol{Y}$. In comparison, we may view equation 6:

$$\max_{\boldsymbol{W}} \frac{1}{4} \left\| \boldsymbol{W}^* \boldsymbol{Y} \right\|_4^4 \quad \text{subject to } \boldsymbol{W} \in \mathsf{St}(n, k; \mathbb{R})$$

as finding a direction, or a $k$-dimensional subspace, in $\mathrm{row}(\boldsymbol{Y})$ where the projection of $\boldsymbol{Y}$ has the largest $\ell^4$-norm. For instance, finding the direction with the largest $\ell^4$-norm over the unit sphere (equation 6) can be viewed as calculating the induced $\left\| \cdot \right\|_{2,4}$ norm of matrix $\boldsymbol{Y}$: $\left\| \boldsymbol{Y} \right\|_{2,4} \doteq \max_{\boldsymbol{a} \in \mathbb{S}^{n-1}} \left\| \boldsymbol{a}^* \boldsymbol{Y} \right\|_4$.

**Relation with the Statistical Interpretation of PCA.**   View each column $\boldsymbol{y}_j, j \in [p]$ of data matrix $\boldsymbol{Y} \in \mathbb{R}^{n \times p}$ as an $n$ dimensional random vector that is i.i.d. drawn from a distribution of random variable $\boldsymbol{y}$ and let $\boldsymbol{Y}_c$ denote the centered $\boldsymbol{Y}$: $\boldsymbol{Y}_c \doteq \boldsymbol{Y}[\boldsymbol{I} - \frac{1}{p}\boldsymbol{1}\boldsymbol{1}^*]$, where $\boldsymbol{1} \in \mathbb{R}^p$ is a vector of all 1's. Then, finding the top $k$ principal components of $\boldsymbol{Y}_c$: $\max_{\boldsymbol{W} \in \mathsf{St}(n,k;\mathbb{R})} \frac{1}{2} \left\| \boldsymbol{W}^* \boldsymbol{Y}_c \right\|_2^2$ is to find $k$ *uncorrelated* projections of $\boldsymbol{y} \in \mathbb{R}^n$ that has the top $k$ sample variance (i.e. $2^{\mathrm{nd}}$ order moment) (Jolliffe, 2011; Helwig, 2017). Similar to PCA, the $\ell^4$-norm maximization of centered data matrix $\boldsymbol{Y}_c$: $\max_{\boldsymbol{W} \in \mathsf{St}(n,k;\mathbb{R})} \frac{1}{4} \left\| \boldsymbol{W}^* \boldsymbol{Y}_c \right\|_4^4$ can be viewed as finding $k$ *uncorrelated* projections of $\boldsymbol{y}$ that have the top $k$ sample $4^{\mathrm{th}}$ order moment, whose statistical meaning is better revealed below.

**Relation with ICA and Nonnormality.**   The $\ell^4$-norm maximization over the Stiefel manifold is strongly related to finding the maximal or minimal *kurtosis* in *Independent Component Analysis* (ICA) (Hyvärinen & Oja, 1997; 2000): In order to identify one component of a given random vector $\boldsymbol{y} \in \mathbb{R}^n$, ICA aims to find a unit vector (a direction) $\boldsymbol{w} \in \mathbb{S}^{n-1}$ that maximizes or minimizes the kurtosis of $\boldsymbol{w}^* \boldsymbol{y}$, defined as:

$$\mathrm{kurt}(\boldsymbol{w}^* \boldsymbol{y}) = \mathbb{E} \left( \boldsymbol{w}^* \boldsymbol{y} \right)^4 - 3 \left\| \boldsymbol{w} \right\|_2^4. \tag{8}$$

Kurtosis is widely used for evaluating the *nonnormality* of a random variable; see DeCarlo (1997); Hyvärinen & Oja (1997; 2000). According to Huber (1985), the nonnormality of data carries "ab-normal" hence interesting information in real data for many applications (e.g. Lee et al. (2003); Cain

et al. (2017)). Thus, extracting the $4^{\text{th}}$ order moment helps understand such statistics of real datasets (Hyvärinen et al., 2009) and even their topology (Carlsson, 2009). One may also find that the $\ell^4$-maximization based dictionary learning formulation is similar to maximizing kurtosis (equation 8) with spherical constraint $\boldsymbol{w} \in \mathbb{S}^{n-1}$, in fact, these two formulations are exactly the same if one only wants to find one column of the dictionary. Intuitively, such coincidence occurs due to the fact that maximizing $\ell^4$-norm or kurtosis have the same effect— they both promote the "spikiness" (Zhang et al. (2018); Li & Bresler (2018)) (or "peak" (DeCarlo, 1997)) of a distribution. More rigorous analysis regarding the similarity between $\ell^4$-based dictionary learning and ICA is beyond the scope of this paper and we leave them for future research.

## 2.2 FIXED-POINT STYLE ALGORITHMS

In optimization, the $\ell^4$-norm maximization in equation 6 over the Stiefel manifold $\mathsf{St}(k, n; \mathbb{R})$ is a special type of nonconvex optimization problem – convex maximization over a compact set. Although Journée et al. (2010); Zhai et al. (2019b) have shown that the MSP algorithm is guaranteed to find critical points, the experiments in Zhai et al. (2019b) suggest that the MSP algorithm finds global maxima of the $\ell^4$-norm efficiently and effectively. For better understanding, in this section we illustrate some striking similarities between the MSP algorithm and the "power-iteration" type algorithms for solving PCA as well as ICA.

**Fixed-point Perspective of Power Iteration.** For a general data matrix $\boldsymbol{Y} \in \mathbb{R}^{n \times p}$, finding the top singular value of $\boldsymbol{Y}$ is equivalent to solving the following optimization problem:

$$\max_{\boldsymbol{w}} \varphi(\boldsymbol{w}) \doteq \frac{1}{2} \|\boldsymbol{w}^* \boldsymbol{Y}\|_2^2 \quad \text{subject to} \ \ \boldsymbol{w} \in \mathbb{S}^{n-1}. \tag{9}$$

For this constrained optimization, the Lagrangian multiplier method gives the necessary condition: $\nabla_{\boldsymbol{w}} \varphi(\boldsymbol{w}) = \boldsymbol{Y} \boldsymbol{Y}^* \boldsymbol{w} = \lambda \boldsymbol{w}$, similar to equation 4. If we restrict this condition onto the sphere, we obtain the fixed point condition $\boldsymbol{w} = \mathcal{P}_{\mathbb{S}^{n-1}} [\nabla_{\boldsymbol{w}} \varphi(\boldsymbol{w})]$. The classic power-iteration method

$$\boldsymbol{w}_{t+1} = \mathcal{P}_{\mathbb{S}^{n-1}} [\nabla_{\boldsymbol{w}} \varphi(\boldsymbol{w}_t)] = \frac{\boldsymbol{Y} \boldsymbol{Y}^* \boldsymbol{w}_t}{\|\boldsymbol{Y} \boldsymbol{Y}^* \boldsymbol{w}_t\|_2}, \tag{10}$$

is precisely computing this fixed point, which is arguably the most efficient and widely used algorithm to solve equation 9, for PCA (or computing SVD of $\boldsymbol{Y}$).

**Fixed-point Perspective of FastICA.** In order to maximize (or minimize) the kurtosis over $\mathbb{S}^{n-1}$,

$$\max_{\boldsymbol{w}} \psi(\boldsymbol{w}) \doteq \frac{1}{4} \text{kurt}[\boldsymbol{w}^* \boldsymbol{y}] = \frac{1}{4} \mathbb{E} [\boldsymbol{w}^* \boldsymbol{y}]^4 - \frac{3}{4} \|\boldsymbol{w}\|_2^4 \quad \text{subject to} \ \ \boldsymbol{w} \in \mathbb{S}^{n-1}, \tag{11}$$

Hyvärinen & Oja (1997) has proposed the following fixed-point type iteration:

$$\boldsymbol{w}_{t+1} = \mathcal{P}_{\mathbb{S}^{n-1}} [\nabla_{\boldsymbol{w}} \psi(\boldsymbol{w}_t)] = \frac{\mathbb{E} \left[ \boldsymbol{y} \left( \boldsymbol{y}^* \boldsymbol{w}_t \right)^3 \right] - 3 \|\boldsymbol{w}_t\|_2^2 \boldsymbol{w}_t}{\left\| \mathbb{E} \left[ \boldsymbol{y} \left( \boldsymbol{y}^* \boldsymbol{w}_t \right)^3 \right] - 3 \|\boldsymbol{w}_t\|_2^2 \boldsymbol{w}_t \right\|_2}, \tag{12}$$

which enjoys cubic (at least quadratic) rate of convergence, under the ICA model assumption.

**Fixed-point Perspective of MSP.** For the $\ell^4$-norm maximization program:

$$\max_{\boldsymbol{W}} \phi(\boldsymbol{W}) \doteq \frac{1}{4} \|\boldsymbol{W}^* \boldsymbol{Y}\|_4^4 \quad \text{subject to} \ \ \boldsymbol{W} \in \mathsf{St}(n, k; \mathbb{R}),$$

through a similar derivation to that in Section 1, one can show that the MSP iteration in equation 5 for the orthogonal group generalizes to the Stiefel manifold case as:

$$\boldsymbol{W}_{t+1} = \mathcal{P}_{\mathsf{St}(n,k;\mathbb{R})} [\nabla_{\boldsymbol{W}} \phi(\boldsymbol{W}_t)] = \boldsymbol{U}_t \boldsymbol{V}_t^*, \tag{13}$$

where $\boldsymbol{U}_t \boldsymbol{\Sigma}_t \boldsymbol{V}_t^* = \text{SVD}[\boldsymbol{Y}(\boldsymbol{Y}^* \boldsymbol{W}_t)^{\circ 3}]$. The above iteration has the same nature as the power iteration in equation 10 and equation 12, since they all solve a fixed-point type problem, by projecting gradient of the objective function $\nabla \varphi(\cdot), \nabla \phi(\cdot), \nabla \psi(\cdot)$ onto the constraint manifold $\mathbb{S}^{n-1}$ and $\mathsf{St}(n, k; \mathbb{R})$, respectively. Table 1 summarizes these striking similarities.

|  | Objectives | Constraint Sets | Algorithms |
|---|---|---|---|
| Power Iteration | $\varphi(\boldsymbol{w}) \doteq \frac{1}{2}\|\boldsymbol{w}^*\boldsymbol{Y}\|_2^2$ | $\boldsymbol{w} \in \mathbb{S}^{n-1}$ | $\boldsymbol{w}_{t+1} = \mathcal{P}_{\mathbb{S}^{n-1}}\left[\nabla_{\boldsymbol{w}}\varphi(\boldsymbol{w}_t)\right]$ |
| FastICA | $\psi(\boldsymbol{w}) \doteq \frac{1}{4}\mathrm{kurt}[\boldsymbol{w}^*\boldsymbol{y}]$ | $\boldsymbol{w} \in \mathbb{S}^{n-1}$ | $\boldsymbol{w}_{t+1} = \mathcal{P}_{\mathbb{S}^{n-1}}\left[\nabla_{\boldsymbol{w}}\psi(\boldsymbol{w}_t)\right]$ |
| MSP | $\phi(\boldsymbol{W}) \doteq \frac{1}{4}\|\boldsymbol{W}^*\boldsymbol{Y}\|_4^4$ | $\boldsymbol{W} \in \mathsf{St}(n,k;\mathbb{R})$ | $\boldsymbol{W}_{t+1} = \mathcal{P}_{\mathsf{St}(n,k;\mathbb{R})}\left[\nabla_{\boldsymbol{W}}\phi(\boldsymbol{W}_t)\right]$ |

Table 1: Similarities among fixed-point algorithms for Power Iteration, FastICA, and MSP.

## 3 STABILITY AND ROBUSTNESS OF $\ell^4$-NORM MAXIMIZATION

Even though the MSP algorithm for $\ell^4$-norm maximization is similar to power-iteration for PCA, in real applications, PCA often requires modifications to improve its robustness (Candès et al., 2011; Xu et al., 2010; 2012). In this section, we want to examine the stability and robustness of the $\ell^4$-maximization for different types of imperfect measurement models: small noise, outliers, and sparse corruptions of large magnitude.

### 3.1 DIFFERENT MODELS FOR IMPERFECT MEASUREMENTS

We adopt the same Bernoulli-Gaussian model as in prior works (Spielman et al., 2012; Sun et al., 2015; Bai et al., 2018; Zhai et al., 2019b) to test the stability and robustness of the $\ell^4$-maximization framework. Assume our clean observation matrix $\boldsymbol{Y} \in \mathbb{R}^{n \times p}$ is produced by the product of a ground truth orthogonal dictionary $\boldsymbol{D}_o$ and a Bernoulli-Gaussian matrix $\boldsymbol{X}_o \in \mathbb{R}^{n \times p}$:

$$\boldsymbol{Y} = \boldsymbol{D}_o\boldsymbol{X}_o, \quad \boldsymbol{D}_o \in \mathsf{O}(n;\mathbb{R}), \ \{\boldsymbol{X}_o\}_{i,j} \sim_{iid} \mathrm{BG}(\theta). \tag{14}$$

Now let us assume we only observe different types of imperfect measurements of $\boldsymbol{Y}$:

**Noisy Measurements:** $\boldsymbol{Y}_N := \boldsymbol{Y} + \boldsymbol{G}$, where $\boldsymbol{G} \in \mathbb{R}^{n \times p}$ is matrix that satisfies $g_{i,j} \sim_{iid} \mathcal{N}(0, \eta^2)$ and $\eta > 0$ controls the variance of the noise.

**Measurements with Outliers:** $\boldsymbol{Y}_O := [\boldsymbol{Y}, \boldsymbol{G}']$, where $\boldsymbol{Y}_O$ contains extra columns $(\boldsymbol{G}' \in \mathbb{R}^{n \times \tau p})$[6] generated from an independent Gaussian process $g'_{i,j} \sim_{iid} \mathcal{N}(0,1)$. Here, $\tau$ controls the portion of the outliers w.r.t. the clean data size $p$.

**Measurements with Sparse Corruptions:** $\boldsymbol{Y}_C := \boldsymbol{Y} + \sigma\boldsymbol{B} \circ \boldsymbol{S}$, where $\sigma > 0$ controls the scale of corrupting entries,[7] $\boldsymbol{B} \in \mathbb{R}^{n \times p}$ is a Bernoulli matrix so $b_{i,j} \sim_{iid} \mathrm{Ber}(\beta)$ with $\beta \in (0,1)$ controlling the ratio of the sparse corruptions, and $\boldsymbol{S} \in \mathbb{R}^{n \times p}$ has entries $s_{i,j}$ drawn i.i.d. from a *Rademacher* distribution:

$$s_{i,j} = \begin{cases} 1 & \text{with probability } 1/2 \\ -1 & \text{with probability } 1/2 \end{cases}. \tag{15}$$

### 3.2 STATISTICAL ANALYSIS AND JUSTIFICATION

The analysis for the stability and robustness of the $\ell^4$-norm maximization follows similar statistical analysis techniques in Zhai et al. (2019b) to show that the global maximum of

$$\boldsymbol{W}_\star \in \arg\max_{\boldsymbol{W}} \mathbb{E}\|\boldsymbol{W}^*\boldsymbol{Y}_\diamond\|_4^4, \quad \text{subject to } \boldsymbol{W} \in \mathsf{O}(n;\mathbb{R}) \tag{16}$$

satisfies $\boldsymbol{W}_\star^*\boldsymbol{D}_o \in \mathrm{SP}(n)$.[8] Note here $\boldsymbol{Y}_\diamond$ denotes different imperfect measurements: noisy $(\boldsymbol{Y}_N)$, with outliers $(\boldsymbol{Y}_O)$, and with sparse corruptions $(\boldsymbol{Y}_C)$. Below we provide the expectation and concentration results for $\|\boldsymbol{W}^*\boldsymbol{Y}_\diamond\|_4^4$ over the data distribution. We show that $\mathbb{E}\|\boldsymbol{W}^*\boldsymbol{Y}_\diamond\|_4^4$ is largely determined by $\|\boldsymbol{W}\boldsymbol{D}_o\|_4^4$, a quantity that indicates a "distance" from $\boldsymbol{W}^*\boldsymbol{D}_o$ to $\mathrm{SP}(n)$. As shown in Lemma 2.3 and Lemma 2.4 in Zhai et al. (2019b), the *only global maximizers* of $\|\boldsymbol{W}^*\boldsymbol{D}_o\|_4^4$ are signed permutation matrices, and $\boldsymbol{W}^*\boldsymbol{D}_o$ converges to a signed permutation matrix as $\|\boldsymbol{W}^*\boldsymbol{D}_o\|_4^4$ reaches its global maximum.

---

[6]In case $\tau p$ is not an integer, we round $\tau p$ to the closest integer.

[7]In our context, $\sigma = 1$ is already corruption of large magnitude, since the variance of the sparse signal is 1.

[8]$\mathrm{SP}(n)$ is the signed permutation group, a group of orthogonal matrices that only contain $0, \pm 1$.

**Proposition 3.1 (Expectation of Objective with Small Noise)** $\forall \theta \in (0, 1)$, let $\boldsymbol{X}_o \in \mathbb{R}^{n \times p}$, $x_{i,j} \sim_{iid} BG(\theta)$. Let $\boldsymbol{D}_o \in \mathsf{O}(n; \mathbb{R})$ be an orthogonal matrix and assume $\boldsymbol{Y} = \boldsymbol{D}_o \boldsymbol{X}_o$. For any orthogonal matrix $\boldsymbol{W} \in \mathsf{O}(n; \mathbb{R})$ and any random Gaussian matrix $\boldsymbol{G} \in \mathbb{R}^{n \times p}$, $g_{i,j} \sim_{iid} \mathcal{N}(0, \eta^2)$ independent of $\boldsymbol{X}_o$, let $\boldsymbol{Y}_N = \boldsymbol{Y} + \boldsymbol{G}$ denote the data with noise. Then the expectation of $\frac{1}{np} \|\boldsymbol{W}^* \boldsymbol{Y}_N\|_4^4$ satisfies:

$$\frac{1}{np} \mathbb{E}_{\boldsymbol{X}_o, \boldsymbol{G}} \|\boldsymbol{W}^* \boldsymbol{Y}_N\|_4^4 = 3\theta(1-\theta) \frac{\|\boldsymbol{W}^* \boldsymbol{D}_o\|_4^4}{n} + C_{\theta, \eta}, \tag{17}$$

where $C_{\theta, \eta}$ is a constant which depends on $\theta$ and $\eta$.

**Proof** See Appendix A.1. ∎

**Theorem 3.2 (Concentration of Objective with Small Noise)** $\forall \theta \in (0, 1)$, let $\boldsymbol{X}_o \in \mathbb{R}^{n \times p}$, $x_{i,j} \sim_{iid} BG(\theta)$. Let $\boldsymbol{D}_o \in \mathsf{O}(n; \mathbb{R})$ be an orthogonal matrix and assume $\boldsymbol{Y} = \boldsymbol{D}_o \boldsymbol{X}_o$. For any orthogonal matrix $\boldsymbol{W} \in \mathsf{O}(n; \mathbb{R})$ and any random Gaussian matrix $\boldsymbol{G} \in \mathbb{R}^{n \times p}$, $g_{i,j} \sim_{iid} \mathcal{N}(0, \eta^2)$ independent of $\boldsymbol{X}_o$, let $\boldsymbol{Y}_N = \boldsymbol{Y} + \boldsymbol{G}$ denote the input with noise, then:

$$\mathbb{P}\left( \sup_{\boldsymbol{W} \in \mathsf{O}(n;\mathbb{R})} \frac{1}{np} \left| \|\boldsymbol{W}^* \boldsymbol{Y}_N\|_4^4 - \mathbb{E} \|\boldsymbol{W}^* \boldsymbol{Y}_N\|_4^4 \right| \geq \delta \right) < \frac{1}{p}, \tag{18}$$

when $p = \Omega\left((1 + \eta^2)^4 n^2 \ln n / \delta^2\right)$.

**Proof** See Appendix A.2. ∎

**Proposition 3.3 (Expectation of Objective with Outliers)** $\forall \theta \in (0, 1)$, let $\boldsymbol{X}_o \in \mathbb{R}^{n \times p}$, $x_{i,j} \sim_{iid} BG(\theta)$. Let $\boldsymbol{D}_o \in \mathsf{O}(n; \mathbb{R})$ be an orthogonal matrix and assume $\boldsymbol{Y} = \boldsymbol{D}_o \boldsymbol{X}_o$. For any orthogonal matrix $\boldsymbol{W} \in \mathsf{O}(n; \mathbb{R})$ and any random Gaussian matrix $\boldsymbol{G}' \in \mathbb{R}^{n \times \tau p}$, $g'_{i,j} \sim_{iid} \mathcal{N}(0, 1)$ independent of $\boldsymbol{X}_o$, let $\boldsymbol{Y}_O = [\boldsymbol{Y}, \boldsymbol{G}']$ denote the data with outliers $\boldsymbol{G}'$. Then the expectation of $\frac{1}{np} \|\boldsymbol{W}^* \boldsymbol{Y}_O\|_4^4$ satisfies:

$$\frac{1}{np} \mathbb{E}_{\boldsymbol{X}_o, \boldsymbol{G}'} \|\boldsymbol{W}^* \boldsymbol{Y}_O\|_4^4 = 3\theta(1-\theta) \frac{\|\boldsymbol{W}^* \boldsymbol{D}_o\|_4^4}{n} + C_{\theta, \tau}, \tag{19}$$

where $C_{\theta, \tau}$ is a constant depends on $\theta, \tau$.

**Proof** See Appendix A.3 ∎

**Theorem 3.4 (Concentration of Objective with Outliers)** $\forall \theta \in (0, 1)$, let $\boldsymbol{X}_o \in \mathbb{R}^{n \times p}$, $x_{i,j} \sim_{iid} BG(\theta)$. Let $\boldsymbol{D}_o \in \mathsf{O}(n; \mathbb{R})$ be an orthogonal matrix and assume $\boldsymbol{Y} = \boldsymbol{D}_o \boldsymbol{X}_o$. For any orthogonal matrix $\boldsymbol{W} \in \mathsf{O}(n; \mathbb{R})$ and any random Gaussian matrix $\boldsymbol{G}' \in \mathbb{R}^{n \times \tau p}$, $g'_{i,j} \sim_{iid} \mathcal{N}(0, 1)$ independent of $\boldsymbol{X}_o$, let $\boldsymbol{Y}_O = [\boldsymbol{Y}, \boldsymbol{G}']$ denote the input with outlier $\boldsymbol{G}'$, then:

$$\mathbb{P}\left( \sup_{\boldsymbol{W} \in \mathsf{O}(n;\mathbb{R})} \frac{1}{np} \left| \|\boldsymbol{W}^* \boldsymbol{Y}_O\|_4^4 - \mathbb{E} \|\boldsymbol{W}^* \boldsymbol{Y}_O\|_4^4 \right| \geq \delta \right) < \frac{1}{p}, \tag{20}$$

when $p = \Omega(\tau^2 n^2 \ln n / \delta^2)$.

**Proof** See Appendix A.4. ∎

In the above results, Proposition 3.1 and 3.3 reveal that both normalized $\frac{1}{np} \mathbb{E} \|\boldsymbol{W}^* \boldsymbol{Y}_N\|_4^4$, $\frac{1}{np} \mathbb{E} \|\boldsymbol{W}^* \boldsymbol{Y}_O\|_4^4$ are only determined by $\|\boldsymbol{W}^* \boldsymbol{D}_o\|_4^4$. Moreover, as shown in Theorem 3.2 and 3.4, when $p$ is large enough ($p = \Omega((1 + \eta^2)^4 n^2 \ln n)$, $\Omega(\tau^2 n^2 \ln n)$ respectively), both $\frac{1}{np} \|\boldsymbol{W}^* \boldsymbol{Y}_N\|_4^4$, $\frac{1}{np} \|\boldsymbol{W}^* \boldsymbol{Y}_O\|_4^4$ concentrate around their expectation with high probability. Therefore, the $\ell^4$-norm maximization formulation $\|\boldsymbol{W}^* \boldsymbol{Y}_\diamond\|_4^4$ for dictionary learning is insensitive to dense Gaussian noise and robust to Gaussian outliers.[9]

---

[9] In the outlier case, Proposition 3.3 and Theorem 3.4 can be generalized to any rotation invariant distributions, e.g., uniform distribution on the sphere. More details regarding rotation invariant distributions can be found in Chapter 4 of Bryc (2012).

**Proposition 3.5 (Expectation of Objective with Sparse Corruptions)** $\forall \theta \in (0,1)$, let $\boldsymbol{X}_o \in \mathbb{R}^{n \times p}$, $x_{i,j} \sim_{iid} BG(\theta)$. Let $\boldsymbol{D}_o \in \mathsf{O}(n; \mathbb{R})$ be an orthogonal matrix and assume $\boldsymbol{Y} = \boldsymbol{D}_o \boldsymbol{X}_o$. For any orthogonal matrix $\boldsymbol{W} \in \mathsf{O}(n; \mathbb{R})$ and any random Bernoulli matrix $\boldsymbol{B} \in \mathbb{R}^{n \times p}, b_{i,j} \sim_{iid} Ber(\beta)$ independent of $\boldsymbol{X}_o$, let $\boldsymbol{Y}_C = \boldsymbol{Y} + \sigma \boldsymbol{B} \circ \boldsymbol{S}$ denote the data with sparse corruptions, and $\boldsymbol{S} \in \mathbb{R}^{n \times p}$ is defined in equation 15. Then the expectation of $\frac{1}{np} \|\boldsymbol{W}^* \boldsymbol{Y}_C\|_4^4$ satisfies:

$$\frac{1}{np} \mathbb{E}_{\boldsymbol{X}_o, \boldsymbol{B}, \boldsymbol{S}} \|\boldsymbol{W}^* \boldsymbol{Y}_C\|_4^4 = 3\theta(1-\theta) \frac{\|\boldsymbol{W}^* \boldsymbol{D}_o\|_4^4}{n} + \sigma^4 \beta(1-3\beta) \frac{\|\boldsymbol{W}\|_4^4}{n} + C_{\theta, \sigma, \beta}, \qquad (21)$$

where $C_{\theta, \sigma, \beta}$ is a constant depending on $\theta, \sigma$ and $\beta$.

**Proof** See Appendix A.5. ∎

**Theorem 3.6 (Concentration of Objective with Sparse Corruptions)** $\forall \theta \in (0,1)$, let $\boldsymbol{X}_o \in \mathbb{R}^{n \times p}$, $x_{i,j} \sim_{iid} BG(\theta)$. Let $\boldsymbol{D}_o \in \mathsf{O}(n; \mathbb{R})$ be an orthogonal matrix and assume $\boldsymbol{Y} = \boldsymbol{D}_o \boldsymbol{X}_o$. For any orthogonal matrix $\boldsymbol{W} \in \mathsf{O}(n; \mathbb{R})$ and any random Bernoulli matrix $\boldsymbol{B} \in \mathbb{R}^{n \times p}, b_{i,j} \sim_{iid} Ber(\beta)$ independent of $\boldsymbol{X}_o$, let $\boldsymbol{Y}_C = \boldsymbol{Y} + \sigma \boldsymbol{B} \circ \boldsymbol{S}$ denote the input with sparse corruptions, and $\boldsymbol{S} \in \mathbb{R}^{n \times p}$ is defined in equation 15, then:

$$\mathbb{P}\left(\sup_{\boldsymbol{W} \in \mathsf{O}(n; \mathbb{R})} \frac{1}{np} \left| \|\boldsymbol{W}^* \boldsymbol{Y}_C\|_4^4 - \mathbb{E}\|\boldsymbol{W}^* \boldsymbol{Y}_C\|_4^4 \right| \geq \delta \right) < \frac{1}{p}, \qquad (22)$$

when $p = \Omega\left(\sigma^8 \beta n^2 \ln n / \delta^2\right)$.

**Proof** See Appendix A.6. ∎

Unlike the cases with noise and outliers, Proposition 3.5 and Theorem 3.6 indicate that $\frac{1}{np} \mathbb{E}\|\boldsymbol{W}^* \boldsymbol{Y}_C\|_4^4$ depends on both $\|\boldsymbol{W}^* \boldsymbol{D}_o\|_4^4$ and $\|\boldsymbol{W}\|_4^4$; when $p = \Omega(\sigma^8 \beta n^2 \ln n)$, the objective $\frac{1}{np} \|\boldsymbol{W}^* \boldsymbol{Y}_C\|_4^4$ concentrates around this expectation with high probability. Nevertheless, when the magnitude of $\sigma^4 \beta(1 - 3\beta)$ is significantly smaller than $3\theta(1 - \theta)$, the landscape of the objective $\|\boldsymbol{W}^* \boldsymbol{Y}_C\|_4^4$ would largely be determined by $\|\boldsymbol{W}^* \boldsymbol{D}_o\|_4^4$ only. As shown in Figure 1, this is indeed the case whenever: (a) the sparsity level $\theta$ of ground truth signal $\boldsymbol{X}_o$, is "reasonably" small (neither diminishing to 0 nor larger than 0.5);(b) $\beta$, the sparsity level of the corruption, is small (smaller than 0.5); (c) $\sigma$, the magnitude of the sparse errors, is not significantly larger than the in-

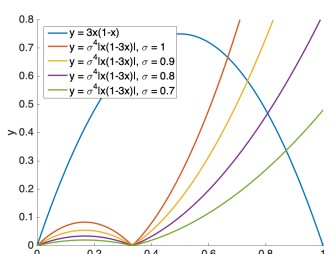

Figure 1: Comparison between $y = 3x(1 - x)$ and $y = \sigma^4 |x(1 - 3x)|$ when $x \in [0, 1]$ with different $\sigma$.

trinsic variance of the sparse signal (the intrinsic variance of the sparse signal Bernoulli-Gaussian model is 1).

Therefore, besides the insensitivity to small noise and robustness to outliers, the $\ell^4$-maximization based dictionary learning also shows resilience to sparse corruptions under reasonable conditions.

## 4 SIMULATIONS AND EXPERIMENTS

### 4.1 QUANTITATIVE EVALUATION: SIMULATIONS ON SYNTHETIC DATA

**Single Trial of MSP.** In this simulation, we run the MSP algorithm from equation 3, using the imperfect measurements $\boldsymbol{Y}_\diamond$ of different models ($\boldsymbol{Y}_N, \boldsymbol{Y}_O, \boldsymbol{Y}_C$). As shown in Figure 2, the normalized value of $\|\boldsymbol{W}^* \boldsymbol{D}_o\|_4^4 / n$ reaches global maximum with all types of inputs when varying the level of noise, outliers, and sparse corruptions. Moreover, as the scale of imperfect measurements increase, Figure 2 shows that (a) the iterations for convergence increases and (b) the final objective value $\|\boldsymbol{W}^* \boldsymbol{D}_o\|_4^4$ decreases almost negligibly. This numerical experiment suggests that the MSP algorithm is able to identify the ground truth orthogonal transformation $\boldsymbol{D}_o$ despite different types of imperfect measurements.

**Phase Transition.** Next, we conduct extensive simulations to study the relation between recovery accuracy and sample size $p$. We run the experiments by increasing the sample size $p$ w.r.t. the scale of imperfect measurements $\eta, \tau, \beta$, respectively. As shown in Figure 3, the MSP algorithm (3)

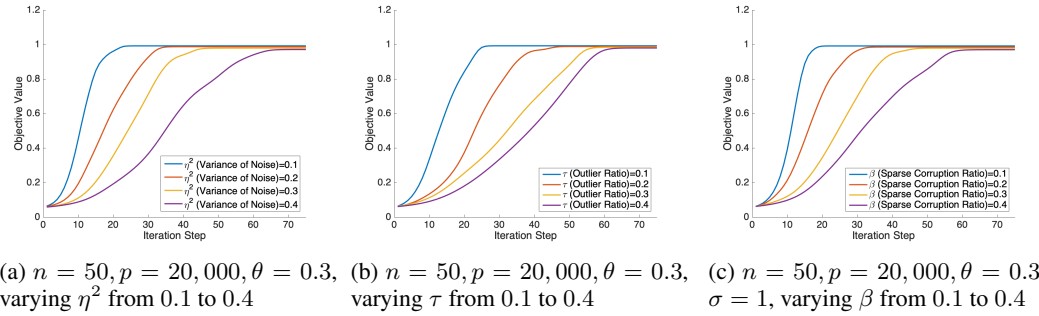

(a) $n = 50, p = 20,000, \theta = 0.3$, varying $\eta^2$ from 0.1 to 0.4

(b) $n = 50, p = 20,000, \theta = 0.3$, varying $\tau$ from 0.1 to 0.4

(c) $n = 50, p = 20,000, \theta = 0.3$, $\sigma = 1$, varying $\beta$ from 0.1 to 0.4

Figure 2: Normalized $\|\boldsymbol{W}^*\boldsymbol{D}_o\|_4^4 / n$ of the MSP algorithm for dictionary learning, using imperfect measurements $\boldsymbol{Y}_N, \boldsymbol{Y}_O, \boldsymbol{Y}_C$, respectively.

demonstrates a clear phase transition behavior w.r.t. noise, outliers, and sparse corruptions. Such phenomena suggest that the algorithm is inherently stable and robust to certain amounts of noise, outliers, and sparse corruptions. The results corroborate our concentration results in Theorem 3.2, 3.4 and 3.6—a larger sample size $p$ increases the accuracy and robustness of the MSP algorithm (3) for all types of nuisances.

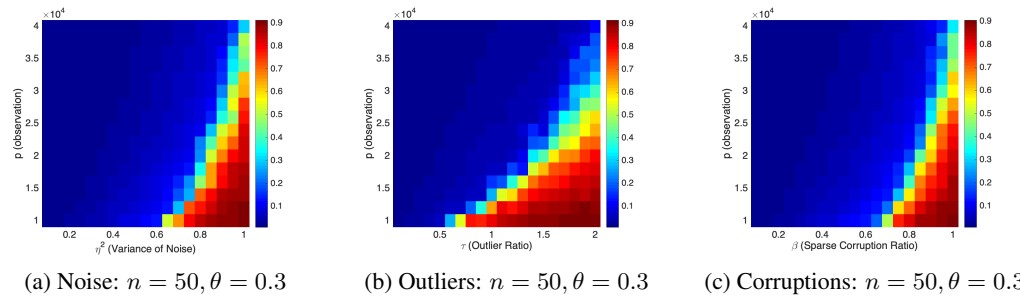

(a) Noise: $n = 50, \theta = 0.3$

(b) Outliers: $n = 50, \theta = 0.3$

(c) Corruptions: $n = 50, \theta = 0.3$

Figure 3: Average normalized error $|1 - \|\boldsymbol{W}^*\boldsymbol{D}_o\|_4^4 / n|$ of 10 random trials for the MSP algorithm: **(a)** Varying sample size $p$ and variance of noise $\eta^2$; **(b)** Varying sample size $p$ and Gaussian Outlier ratio $\tau$; **(c)** Varying sample size $p$ and sparse corruption ratio $\beta$, with fixed $\sigma = 1$.

**Comparison with Prior Arts.** We also compare the MSP algorithm (Zhai et al., 2019b) with previous complete dictionary learning algorithms: Subgradient (SG) (Bai et al., 2018) and Riemannian Trust Region (RTR) methods (Sun et al., 2015). While SG and RTR demonstrate slightly better accuracy in some cases (e.g. small Gaussian noise and Gaussian outliers), both algorithms appear unstable to sparse corruptions. Meanwhile, the MSP algorithm is stable to all imperfect measurement models mentioned in this paper and runs significantly faster than the others.

| | | | Clean | | Noise | | | | Outlier | | | | Corruption | | | |
| | | | | | 0.2 | | 0.4 | | 0.2 | | 0.4 | | 0.2 | | 0.4 | |
| $n$ | $p$ | Alg. | Error | Time | Error | Time | Error | Time | Error | Time | Error | Time | Error | Time | Error | Time |
|---|---|---|---|---|---|---|---|---|---|---|---|---|---|---|---|---|---|
| 25 | 10k | MSP | 0.34% | **1.14s** | 0.45% | **1.11s** | **0.99%** | **1.12s** | 1.11% | **1.29s** | 1.82% | **1.45s** | **1.27%** | **1.05s** | **2.85%** | **1.03s** |
| | | SG | **0.00%** | 8.00m | 5.54% | 19.3m | 25.7% | 26.4m | **0.00%** | 9.62m | **0.00%** | 11.1m | 87.0% | 16.6m | 88.4% | 34.2m |
| | | RTR | **0.00%** | 3.28m | **0.03%** | 17.0m | 1.34% | 20.2m | **0.00%** | 4.63m | **0.00%** | 5.87m | 3.20% | 18.0m | 33.9% | 16.3m |
| 50 | 20k | MSP | 0.34% | **4.82s** | 0.47% | **4.65s** | 1.02% | **5.44s** | 1.15% | **5.31s** | 2.01% | **6.45s** | 1.33% | **4.80s** | 3.04% | **4.41s** |
| | | SG | **0.00%** | 1.34h | N/A | >2h | N/A | >2h | 3.42% | 1.40h | **0.00%** | 1.81h | N/A | >2h | N/A | >2h |
| | | RTR | **0.00%** | 23.0m | **0.04%** | 1.57h | 1.65% | 1.38h | **0.00%** | 30.7m | **0.00%** | 41.8m | 2.17% | 1.25h | 82.57% | 1.29h |

Table 2: Comparison of the MSP algorithm (Zhai et al., 2019b) with prior complete dictionary learning algorithms: Subgradient method (Bai et al., 2018) and Riemannian Trust Region (Sun et al., 2015) methods in different models with fixed ground truth sparsity $\theta = 0.3$. Note that SG only learns a unit vector each time and does not guarantee orthogonality; we therefore project the dictionary learned to $\mathsf{O}(n; \mathbb{R})$ for fair comparison.

## 4.2 QUALITATIVE EVALUATION: EXPERIMENTS ON REAL IMAGES AND PATCHES

Besides simulations, we also conduct extensive experiments to verify the stability and robustness of the MSP algorithm with real imagery data, at both image level and patch level. Throughout these experiments, rather than visualize all bases, we routinely show the top bases learned—heuristically, the top bases are those with the largest coefficients (here, in terms of $\ell^1$-norm).

**Image Level.** At the image level, we first vectorize all 60,000 images in the MNIST dataset (Le-Cun et al., 1998) into a single matrix $Y \in \mathbb{R}^{784 \times 60,000}$, then create imperfect measurements based on models specified in Section 3: $Y_N$ (MNIST with noise), $Y_O$ (MNIST with outliers), $Y_C$ (MNIST with sparse corruptions). We run the MSP algorithm 3 with $Y, Y_N, Y_C, Y_O$ and compare the learned bases. Figure 4(a), (c), (e), and (g) show examples of $Y, Y_N, Y_O$, and $Y_C$, and Figure 4(b), (d), (f), and (h) show top 10 bases learned from $Y, Y_N, Y_C, Y_O$, respectively. Despite that we use different types of imperfect measurements of MNIST, the top bases learned from MSP algorithm 3 are very much the same.[10] This result corroborates our analysis: the $\ell^4$-maximization and the MSP algorithm is inherently insensitive to noise, robust to outliers, and resilient to sparse corruptions.

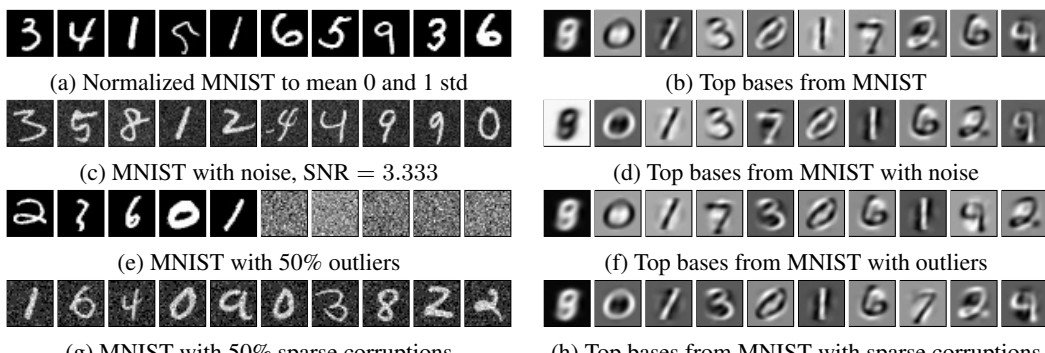

(a) Normalized MNIST to mean 0 and 1 std      (b) Top bases from MNIST

(c) MNIST with noise, SNR $= 3.333$      (d) Top bases from MNIST with noise

(e) MNIST with 50% outliers      (f) Top bases from MNIST with outliers

(g) MNIST with 50% sparse corruptions      (h) Top bases from MNIST with sparse corruptions

Figure 4: **Left:** Examples of MNIST and its different imperfect measurements. **Right:** Learned bases from MNIST and its different imperfect measurements using the MSP algorithm 3.

**Patch Level.** A classic application of dictionary learning involves learning sparse representations of image patches (Elad & Aharon, 2006; Mairal et al., 2007). In this section, we extend the experiments of Zhai et al. (2019b) to learn patches from gray scale and color images. First, we construct a data matrix $Y$ by vectorizing each $8 \times 8$ patch from the $512 \times 512$ gray scale image, "Barbara" (see Figure 5). We then run the MSP algorithm with 100 iterations on both $Y$ and a noisy version $Y_N$, and the learned top bases are visualized in Figure 5.

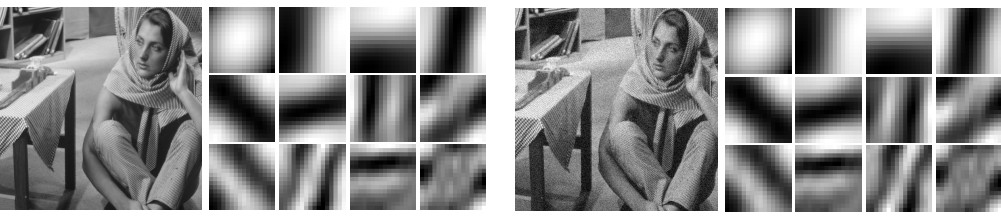

(a) Clean Image and Bases      (b) Noisy Image and Bases(SNR $= 5.87$)

Figure 5: The top 12 bases learned from all $16 \times 16$ patches of Barbara, both with (b) and without (a) noise. The noisy image is produced by adding Gaussian noise to the clean image, resulting in a signal-to-noise ratio (SNR) of $5.87$. We observed a similar effect when using an $8 \times 8$ patch size.

Analogously, we apply the same scheme to a $256 \times 256$ color image, "Duck" (see Figure 6), converting each $8 \times 8 \times 3$ patch into a column vector (in $\mathbb{R}^{192}$) of $Y$. Notice this forces the algorithm to learn bases involving all three channels simultaneously, rather than one at a time. After running the MSP algorithm for 100 iterations, we visualize the top bases learned from both $Y$ and corresponding $Y_N$ in Figure 6.

We next consider the problem of learning a "global dictionary" (Mairal et al., 2007) for patches from many different images. To construct our data matrix $Y$, we randomly sample 100,000 $8 \times 8 \times 3$

---

[10]Bases with opposite intensity are considered as the same base, since they only differ by a sign.

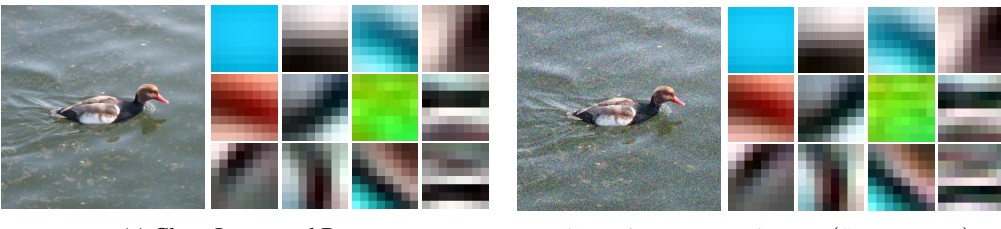

(a) Clean Image and Bases                    (b) Noisy Image and Bases (SNR = 6.56)

Figure 6: The top 12 bases learned from all $8 \times 8 \times 3$ color patches of the clean and noisy image, respectively. Here, the SNR of the noisy image is 6.56.

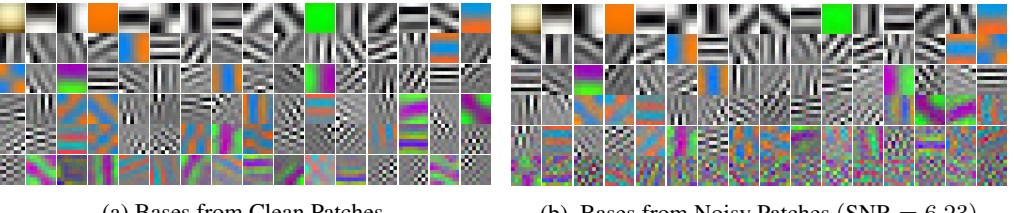

(a) Bases from Clean Patches                    (b) Bases from Noisy Patches (SNR = 6.23)

Figure 7: Top half (96) bases learned from $100,000$ random $8 \times 8 \times 3$ patches sampled from CIFAR-10, before and after adding Gaussian noise, with SNR 6.23.

patches from the CIFAR-10 data-set (Krizhevsky et al., 2009). A noisy data matrix $\boldsymbol{Y}_N$, is then generated by adding Gaussian noise. Again, we apply the MSP algorithm with 200 iterations to learn 192 bases and visualize the results in Figure 7. We leave the experiments of CIFAR-10 with outliers and sparse corruptions in the Appendix due to limited space.

In each of these experiments, the top bases in the learned dictionary remain relatively unchanged with the addition of noise. To quantify this similarity, we take the top bases from the noisy dictionary and find the closest top clean base for each. If the bases are nearly identical, then the inner product of each of these pairs should be close to 1. Table 3 reports the statistics.

|          | Minimum | Lower Quartile | Median | Upper Quartile | Maximum |
|----------|---------|----------------|--------|----------------|---------|
| Barbara  | 0.3048  | 0.8471         | 0.9941 | 0.9993         | 1.0000  |
| Duck     | 0.2510  | 0.9782         | 0.9891 | 0.9971         | 1.0000  |
| CIFAR-10 | 0.5147  | 0.7203         | 0.9892 | 0.9998         | 1.0000  |

Table 3: Statistics about the inner products between the top 20 noisy bases and their corresponding closest top-20 clean bases.

## 5 CONCLUSION AND DISCUSSIONS

In this paper, we find the $\ell^4$-norm maximization based dictionary learning and corresponding MSP algorithm introduced by Zhai et al. (2019b) have strong geometric and statistical connections to classic data analysis methods PCA and ICA. These connections seem to be the reason why they all admit similar simple and efficient algorithms. Empirically, we have observed that $\ell^4$-norm maximization is surprisingly insensitive to noise, robust to outliers, and resilient to sparse corruptions. Moreover, such empirical observations corroborate our concentration analysis—larger data samples ($p$) improve the stability and robustness of the $\ell^4$-maximization based dictionary learning.

From experiments on real images, we observed that top bases learned are rather stable but tail bases can be less stable (see Figure 10 in Appendix C). We conjecture this phenomenon happens due to the fact that real images generally do not follow the uniformly sparse Bernoulli Gaussian model (of equation 14). Generalizing dictionary learning to non-uniformly sparsely generated data would be a good topic for future study. Finally, note that Bai et al. (2018) has established subgradient descent can find global optimal solutions for dictionary learning, an interesting theoretical guarantee since dictionary learning problems do not satisfy the standard structural (i.e. generalized convexity) assumptions that guarantee global convergence (Ben-Tal & Nemirovski, 2001; Zhou et al., 2017a; Nesterov, 2013; Zhou et al., 2017b; Bubeck et al., 2015; Zhou et al., 2017c; 2018; Ma et al., 2018; Chi et al., 2019). It would be desirable, although highly challenging, to establish similar global convergence guarantees for our MSP algorithm. We leave that for future work.

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

## A  PROOF OF SECTION 3

### A.1  PROOF OF PROPOSITION 3.1

**Claim A.1 (Expectation of Objective with Small Noise)** $\forall \theta \in (0,1)$, let $\boldsymbol{X}_o \in \mathbb{R}^{n \times p}$, $x_{i,j} \sim_{iid} BG(\theta)$. Let $\boldsymbol{D}_o \in \mathsf{O}(n; \mathbb{R})$ be an orthogonal matrix and assume $\boldsymbol{Y} = \boldsymbol{D}_o \boldsymbol{X}_o$. For any orthogonal matrix $\boldsymbol{W} \in \mathsf{O}(n; \mathbb{R})$ and any random Gaussian matrix $\boldsymbol{G} \in \mathbb{R}^{n \times p}$, $g_{i,j} \sim_{iid} \mathcal{N}(0, \eta^2)$ independent of $\boldsymbol{X}_o$, let $\boldsymbol{Y}_N = \boldsymbol{Y} + \boldsymbol{G}$ denote the data with noise. Then the expectation of $\frac{1}{np} \|\boldsymbol{W}^* \boldsymbol{Y}_N\|_4^4$ satisfies:

$$\frac{1}{np} \mathbb{E}_{\boldsymbol{X}_o, \boldsymbol{G}} \|\boldsymbol{W}^* \boldsymbol{Y}_N\|_4^4 = 3\theta(1-\theta) \frac{\|\boldsymbol{W}^* \boldsymbol{D}_o\|_4^4}{n} + 3\theta^2 + 6\theta\eta^2 + 3\eta^4. \tag{23}$$

**Proof** Let $\boldsymbol{W}^* \boldsymbol{D}_o = \boldsymbol{M} \in \mathsf{O}(n; \mathbb{R})$, notice that the orthogonal transformation $(\boldsymbol{W}^* \boldsymbol{G})$ of a Gaussian matrix $(\boldsymbol{G})$ is still a Gaussian matrix and satisfies $\{\boldsymbol{W}^* \boldsymbol{G}\}_{i,j} \sim \mathcal{N}(0,1)$, and it is independent of $\boldsymbol{Y}(\boldsymbol{X}_o)$. We abuse the notation a bit let $\boldsymbol{G} = \boldsymbol{W}^* \boldsymbol{G}$ in the following calculation, since $\boldsymbol{W}^* \boldsymbol{G}$ is also a Gaussian matrix independent of $\boldsymbol{X}_o$ and it will not affect the final result.

$$\begin{aligned}
&\mathbb{E}_{\boldsymbol{X}_o, \boldsymbol{G}} \|\boldsymbol{W}^* \boldsymbol{Y}_N\|_4^4 = \mathbb{E}_{\boldsymbol{X}_o, \boldsymbol{G}} \|\boldsymbol{M} \boldsymbol{X}_o + \boldsymbol{G}\|_4^4 \\
&= \sum_{j=1}^p \sum_{i=1}^n \mathbb{E}_{\boldsymbol{X}_o, \boldsymbol{G}} \left( \sum_{k=1}^n m_{i,k} x_{k,j} + g_{i,j} \right)^4 \\
&= \sum_{j=1}^p \sum_{i=1}^n \left\{ \mathbb{E}_{\boldsymbol{X}_o, \boldsymbol{G}} \left( \sum_{k=1}^n m_{i,k} x_{k,j} \right)^4 + 6 \mathbb{E}_{\boldsymbol{X}_o, \boldsymbol{G}} \left[ n_{i,j}^2 \left( \sum_{k=1}^n m_{i,k} x_{k,j} \right)^2 \right] + \mathbb{E}_{\boldsymbol{G}} g_{i,j}^4 \right\} \\
&= \sum_{j=1}^p \sum_{i=1}^n \left\{ \mathbb{E}_{\boldsymbol{X}_o, \boldsymbol{G}} \left( \sum_{k=1}^n m_{i,k} x_{k,j} \right)^4 + 6\eta^2 \mathbb{E}_{\boldsymbol{X}_o} \left( \sum_{k=1}^n m_{i,k} x_{k,j} \right)^2 \right\} + 3np\eta^4 \\
&= \sum_{j=1}^p \sum_{i=1}^n \left\{ \mathbb{E}_{\boldsymbol{X}_o, \boldsymbol{G}} \left( \sum_{k=1}^n m_{i,k} x_{k,j} \right)^4 \right\} + 6np\theta\eta^2 + 3np\eta^4 \\
&= \mathbb{E}_{\boldsymbol{X}_o} \|\boldsymbol{M} \boldsymbol{X}_o\|_4^4 + 6np\theta\eta^2 + 3np\eta^4 \\
&= 3p\theta(1-\theta) \|\boldsymbol{M}\|_4^4 + 3np\theta^2 + 6np\theta\eta^2 + 3np\eta^4,
\end{aligned} \tag{24}$$

therefore,

$$\frac{1}{np} \mathbb{E}_{\boldsymbol{X}_o, \boldsymbol{G}} \|\boldsymbol{W}^* \boldsymbol{Y}_N\|_4^4 = 3\theta(1-\theta) \frac{\|\boldsymbol{W}^* \boldsymbol{D}_o\|_4^4}{n} + 3\theta^2 + 6\theta\eta^2 + 3\eta^4, \tag{25}$$

which completes the proof. ∎

### A.2  PROOF OF THEOREM 3.2

**Claim A.2 (Concentration of Objective with Small Noise)** $\forall \theta \in (0,1)$, let $\boldsymbol{X}_o \in \mathbb{R}^{n \times p}$, $x_{i,j} \sim_{iid} BG(\theta)$. Let $\boldsymbol{D}_o \in \mathsf{O}(n; \mathbb{R})$ be an orthogonal matrix and assume $\boldsymbol{Y} = \boldsymbol{D}_o \boldsymbol{X}_o$. For any orthogonal matrix $\boldsymbol{W} \in \mathsf{O}(n; \mathbb{R})$ and any random Gaussian matrix $\boldsymbol{G} \in \mathbb{R}^{n \times p}$, $g_{i,j} \sim_{iid} \mathcal{N}(0, \eta^2)$ independent of $\boldsymbol{X}_o$, let $\boldsymbol{Y}_N = \boldsymbol{Y} + \boldsymbol{G}$ denote the input with noise, then:

$$\mathbb{P} \left( \sup_{\boldsymbol{W} \in \mathsf{O}(n; \mathbb{R})} \frac{1}{np} \left| \|\boldsymbol{W}^* \boldsymbol{Y}_N\|_4^4 - \mathbb{E} \|\boldsymbol{W}^* \boldsymbol{Y}_N\|_4^4 \right| \geq \delta \right) < \frac{1}{p}, \tag{26}$$

when $p = \Omega\left( (1+\eta^2)^4 n^2 \ln n / \delta^2 \right)$.

**Proof** According to Proposition 3.1, we know that

$$\frac{1}{np} \mathbb{E}_{\boldsymbol{X}_o, \boldsymbol{G}} \|\boldsymbol{W}^* \boldsymbol{Y}_N\|_4^4 = 3\theta(1-\theta) \frac{\|\boldsymbol{W}^* \boldsymbol{D}_o\|_4^4}{n} + 3\theta^2 + 6\theta\eta^2 + 3\eta^4. \tag{27}$$

Moreover, since $\boldsymbol{Y}_N = [\boldsymbol{y}_1 + \boldsymbol{g}_1, \boldsymbol{y}_2 + \boldsymbol{g}_2, \ldots, \boldsymbol{y}_p + \boldsymbol{g}_p]$, whose columns are independent. Let

$$\boldsymbol{Z} = [\boldsymbol{z}_1, \boldsymbol{z}_2, \ldots, \boldsymbol{z}_p] = \boldsymbol{D}_o^* \boldsymbol{Y}_N = [\boldsymbol{x}_1 + \boldsymbol{g}_1, \boldsymbol{x}_2 + \boldsymbol{g}_2, \ldots, \boldsymbol{x}_p + \boldsymbol{g}_p] = \boldsymbol{X}_o + \boldsymbol{G}, \tag{28}$$

note that we abuse the notation a bit by assuming $\boldsymbol{D}_o^* \boldsymbol{G} = \boldsymbol{G}$, since $\boldsymbol{D}_o^* \boldsymbol{G}$ is also a Gaussian matrix independent of $\boldsymbol{X}_o$ and it will not affect the final result. Define function $f_{\boldsymbol{z}}(\boldsymbol{W}) : \mathsf{O}(n; \mathbb{R}) \mapsto \mathbb{R}$ as

$$f_{\boldsymbol{z}}(\boldsymbol{W}) = \|\boldsymbol{W}^* \boldsymbol{z}\|_4^4. \tag{29}$$

Next, we will check assumption 1, 2, and 3 then apply Lemma B.4.

**Assumption 1:** $\mu(n, p)$. Since $z_{i,j} = x_{i,j} + g_{i,j}$, we know that with probability $\theta$, $z \sim \mathcal{N}(0, 1+\eta^2)$ and with probability $1 - \theta$, $z \sim \mathcal{N}(0, 1)$. Hence,

$$\mathbb{P}\left(\max_{i,j} |z_{i,j}| > B\right) \leq 2np\theta \exp\left(-\frac{B^2}{2 + 2\eta^2}\right) + 2np(1 - \theta) \exp\left(-\frac{B^2}{2}\right) = \mu(n, p). \tag{30}$$

Specifically, we set $B = \ln p$, which yields

$$\mu(n, p) = 2np\theta \exp\left(-\frac{(\ln p)^2}{2 + 2\eta^2}\right) + 2np(1 - \theta) \exp\left(-\frac{(\ln p)^2}{2}\right). \tag{31}$$

**Assumption 2: Lipschitz Constant** $L_f$. By Proposition 3.1, $\forall \boldsymbol{W} \in \mathsf{O}(n; \mathbb{R})$, we know that

$$\mathbb{E} f_{\boldsymbol{z}}(\boldsymbol{W}) = 3\theta(1 - \theta) \|\boldsymbol{W}^* \boldsymbol{D}_o\|_4^4 + 3n\theta^2 + 6n\theta\eta^2 + 3n\eta^4. \tag{32}$$

Hence, $\forall \boldsymbol{W}_1, \boldsymbol{W}_2 \in \mathsf{O}(n; \mathbb{R})$, we have

$$\left| \mathbb{E} \|\boldsymbol{W}_1^* \boldsymbol{z}\|_4^4 - \mathbb{E} \|\boldsymbol{W}_1^* \boldsymbol{z}\|_4^4 \right| = 3\theta(1 - \theta) \left| \|\boldsymbol{W}_1^* \boldsymbol{D}_o\|_4^4 - \|\boldsymbol{W}_2^* \boldsymbol{D}_o\|_4^4 \right|, \tag{33}$$

moreover,

$$\left| \sum_{i,j} \left[ (\boldsymbol{W}_1^* \boldsymbol{D}_o)_{i,j}^4 - (\boldsymbol{W}_2^* \boldsymbol{D}_o)_{i,j}^4 \right] \right|$$

$$= \left| \sum_{i,j} \left\{ \left[ (\boldsymbol{W}_1^* \boldsymbol{D}_o)_{i,j}^2 - (\boldsymbol{W}_2^* \boldsymbol{D}_o)_{i,j}^2 \right] \left[ (\boldsymbol{W}_1^* \boldsymbol{D}_o)_{i,j}^2 + (\boldsymbol{W}_2^* \boldsymbol{D}_o)_{i,j}^2 \right] \right\} \right|$$

$$\leq \left\{ \sum_{i,j} \left[ (\boldsymbol{W}_1^* \boldsymbol{D}_o)_{i,j}^2 - (\boldsymbol{W}_2^* \boldsymbol{D}_o)_{i,j}^2 \right]^2 \right\}^{1/2} \left\{ \sum_{i,j} \left[ (\boldsymbol{W}_1^* \boldsymbol{D}_o)_{i,j}^2 + (\boldsymbol{W}_2^* \boldsymbol{D}_o)_{i,j}^2 \right]^2 \right\}^{1/2} \tag{34}$$

$$= \left\| (\boldsymbol{W}_1^* \boldsymbol{D}_o)^{\circ 2} - (\boldsymbol{W}_1^* \boldsymbol{D}_o)^{\circ 2} \right\|_F \left\| (\boldsymbol{W}_1^* \boldsymbol{D}_o)^{\circ 2} + (\boldsymbol{W}_1^* \boldsymbol{D}_o)^{\circ 2} \right\|_F$$

$$\leq \left\| \boldsymbol{W}_1^* \boldsymbol{D}_o - \boldsymbol{W}_2^* \boldsymbol{D}_o \right\|_F \left\| \boldsymbol{W}_1^* \boldsymbol{D}_o + \boldsymbol{W}_2^* \boldsymbol{D}_o \right\|_F \left( \left\| \boldsymbol{W}_1^* \boldsymbol{D}_o \right\|_F^2 + \left\| \boldsymbol{W}_2^* \boldsymbol{D}_o \right\|_F^2 \right)$$

$$\leq 4n^2 \left\| \boldsymbol{W}_1^* \boldsymbol{D}_o - \boldsymbol{W}_2^* \boldsymbol{D}_o \right\|_2 = 4n^2 \left\| \boldsymbol{W}_1 - \boldsymbol{W}_2 \right\|_2.$$

Hence, we know that

$$\left| \mathbb{E} \|\boldsymbol{W}_1^* \boldsymbol{z}\|_4^4 - \mathbb{E} \|\boldsymbol{W}_1^* \boldsymbol{z}\|_4^4 \right| \leq 12n^2\theta(1 - \theta), \tag{35}$$

which yields

$$L_f = 12n^2\theta(1 - \theta). \tag{36}$$

**Assumption 2: Lipschitz Constant** $\bar{L}_f$. Since $f_{\bar{\boldsymbol{z}}}(\boldsymbol{W}) = \|\boldsymbol{W}^* \bar{\boldsymbol{z}}\|_4^4$, we have

$$\left| \|\boldsymbol{W}_1^* \bar{\boldsymbol{z}}\|_4^4 - \|\boldsymbol{W}_2^* \bar{\boldsymbol{z}}\|_4^4 \right| = \left| \sum_i \left\{ \left[ (\boldsymbol{W}_1^* \bar{\boldsymbol{z}})_i^2 - (\boldsymbol{W}_2^* \bar{\boldsymbol{z}})_i^2 \right] \left[ (\boldsymbol{W}_1^* \bar{\boldsymbol{z}})_i^2 + (\boldsymbol{W}_2^* \bar{\boldsymbol{z}})_i^2 \right] \right\} \right|$$

$$\leq \left\{ \sum_i \left[ (\boldsymbol{W}_1^* \bar{\boldsymbol{z}})_i^2 - (\boldsymbol{W}_2^* \bar{\boldsymbol{z}})_i^2 \right]^2 \right\}^{1/2} \left\{ \sum_i \left[ (\boldsymbol{W}_1^* \bar{\boldsymbol{z}})_i^2 + (\boldsymbol{W}_2^* \bar{\boldsymbol{z}})_i^2 \right]^2 \right\}^{1/2} \tag{37}$$

$$= \underbrace{\left\| (\boldsymbol{W}_1^* \bar{\boldsymbol{z}})^{\circ 2} - (\boldsymbol{W}_2^* \bar{\boldsymbol{z}})^{\circ 2} \right\|_2}_{\Gamma_1} \underbrace{\left\| (\boldsymbol{W}_1^* \bar{\boldsymbol{z}})^{\circ 2} + (\boldsymbol{W}_2^* \bar{\boldsymbol{z}})^{\circ 2} \right\|_2}_{\Gamma_2}.$$

For $\Gamma_1$, we have

$$
\begin{aligned}
\Gamma_1 &= \left\| (W_1^* \bar{z})^{\circ 2} - (W_2^* \bar{z})^{\circ 2} \right\|_2 = \left\| (W_1^* \bar{z}) - (W_2^* \bar{z}) \circ (W_1^* \bar{z}) + (W_2^* \bar{z}) \right\|_2 \\
&\leq \| W_1 - W_2 \|_2 \underbrace{\| W_1 + W_2 \|_2}_{\leq 2} \underbrace{\| \bar{z} \|_2^2}_{\leq nB^2} \leq 2nB^2 \| W_1 - W_2 \|_2 .
\end{aligned}
\tag{38}
$$

For $\Gamma_2$, we have

$$
\begin{aligned}
\Gamma_2 &= \left\| (W_1^* \bar{z})^{\circ 2} + (W_2^* \bar{z})^{\circ 2} \right\|_2 \leq \left\| (W_1^* \bar{z})^{\circ 2} \right\|_2 + \left\| (W_2^* \bar{z})^{\circ 2} \right\|_2 \\
&\leq \| W_1^* \bar{z} \|_2^2 + \| W_2^* \bar{z} \|_2^2 \leq 2nB^2 .
\end{aligned}
\tag{39}
$$

Hence, we know that

$$
\left| \| W_1^* \bar{z} \|_4^4 - \| W_2^* \bar{z} \|_4^4 \right| \leq 4n^2 B^4 \| W_1 - W_2 \|_2 ,
\tag{40}
$$

which yields

$$
\bar{L}_f = 4n^2 B^4 .
\tag{41}
$$

**Assumption 3: Upper Bound $R_1$.**    Notice that $\forall W \in \mathsf{O}(n; \mathbb{R})$,

$$
f_{\bar{z}}(W) = \| W^* \bar{z} \|_4^4 = \| W^* \bar{z} \|_2^4 \left\| \frac{W^* \bar{z}}{\| W^* \bar{z} \|_2} \right\|_4^4 \leq \| W^* \bar{z} \|_2^4 = \| \bar{z} \|_2^4 \leq n^2 B^4 = R_1 .
\tag{42}
$$

**Assumption 3: Upper Bound $R_2$.**    Assume the support for $x$ is $\mathcal{S}$, so we know that know that $\forall i \in [n]$,

$$
z_i = \begin{cases} \sqrt{1 + \eta^2} v_i, & \text{if } i \in \mathcal{S}, \\ v_i, & \text{otherwise}, \end{cases}
\tag{43}
$$

where $v \sim \mathcal{N}(0, I)$ is a Gaussian vector. Let $P_{\mathcal{S}} : \mathbb{R}^n \mapsto \mathbb{R}^n$ be a projection that satisfies $\forall q \in \mathbb{R}^n$

$$
\left( P_{\mathcal{S}} q \right)_i = \begin{cases} \sqrt{1 + \eta^2} q_i, & \text{if } i \in \mathcal{S}, \\ q_i, & \text{otherwise}. \end{cases}
\tag{44}
$$

Moreover, $\forall W \in \mathsf{O}(n; \mathbb{R})$, let $w_i$ denote the $i^{\text{th}}$ column vector of $W$, hence

$$
\begin{aligned}
\mathbb{E}[f_z^2(W)] &= \mathbb{E}\left[ \left( \| W^* z \|_4^4 \right)^2 \right] = \mathbb{E}\left[ \sum_{i=1}^{n} \sum_{j=1}^{n} \langle w_i, z \rangle^4 \langle w_j, z \rangle^4 \right] \\
&\leq \mathbb{E}_{\mathcal{S}} \sum_{i=1}^{n} \sum_{j=1}^{n} \mathbb{E}\left[ \langle P_{\mathcal{S}} w_i, v \rangle^4 \langle P_{\mathcal{S}} w_j, v \rangle^4 \right] \leq \mathbb{E}_{\mathcal{S}} \sum_{i=1}^{n} \sum_{j=1}^{n} \left[ \mathbb{E} \langle P_{\mathcal{S}} w_i, v \rangle^8 \mathbb{E} \langle P_{\mathcal{S}} w_j, v \rangle^8 \right]^{\frac{1}{2}} .
\end{aligned}
\tag{45}
$$

Notice that $v \sim \mathcal{N}(0, I)$, therefore $\forall i \in [n]$, we have

$$
\mathbb{E} \langle P_{\mathcal{S}} w_i, v \rangle^8 = 105 \| P_{\mathcal{S}} w_i \|_2^8 ,
\tag{46}
$$

hence

$$
\begin{aligned}
\mathbb{E}_{\mathcal{S}} \sum_{i=1}^{n} \sum_{j=1}^{n} &\left[ \mathbb{E} \langle P_{\mathcal{S}} w_i, v \rangle^8 \mathbb{E} \langle P_{\mathcal{S}} w_j, v \rangle^8 \right]^{\frac{1}{2}} = 105 \mathbb{E}_{\mathcal{S}} \sum_{i=1}^{n} \sum_{j=1}^{n} \| P_{\mathcal{S}} w_i \|_2^4 \| P_{\mathcal{S}} w_i \|_2^4 \\
&= 105 \sum_{i=1}^{n} \sum_{j=1}^{n} \sum_{k_1, k_2, k_3, k_4} \mathbb{E}_{\mathcal{S}} \left[ \prod_{l=1}^{4} (1 + \eta^2 \mathbb{1}_{k_l \in \mathcal{S}}) w_{i,k_1}^2 w_{i,k_2}^2 w_{j,k_3}^2 w_{j,k_4}^2 \right] \\
&\leq 105 (1 + \eta^2)^4 \sum_{i=1}^{n} \sum_{j=1}^{n} \sum_{k_1, k_2, k_3, k_4} [w_{i,k_1}^2 w_{i,k_2}^2 w_{j,k_3}^2 w_{j,k_4}^2] = 105 (1 + \eta^2)^4 n^2 .
\end{aligned}
\tag{47}
$$

Therefore, we can conclude that

$$
\mathbb{E}[f_z^2(W)] \leq 105 (1 + \eta^2)^4 = R_2 .
\tag{48}
$$

**Applying Lemma B.4 for Concentration.**    Now we apply Lemma B.4 with

1.

$$B = \ln p, \quad \mu(n,p) = 2np\theta \exp\left(-\frac{(\ln p)^2}{4}\right) + 2np(1-\theta)\exp\left(-\frac{(\ln p)^2}{2}\right), \quad (49)$$

2.

$$L_f = 12n^2\theta(1-\theta), \quad \bar{L}_f = 4n^2(\ln p)^4, \tag{50}$$

3.

$$R_1 = n^2(\ln p)^4, \quad R_2 = 105(1+\eta^2)^4 n^2, \tag{51}$$

we have

$$
\begin{aligned}
&\mathbb{P}\left(\sup_{\boldsymbol{W}\in\mathsf{O}(n;\mathbb{R})} \frac{1}{np}\left|\sum_{j=1}^p \left[f_{\boldsymbol{z}_j}(\boldsymbol{W}) - \mathbb{E}f_{\boldsymbol{z}_j}(\boldsymbol{W})\right]\right| \geq \delta\right) \\
=&\mathbb{P}\left(\sup_{\boldsymbol{W}\in\mathsf{O}(n;\mathbb{R})} \frac{1}{np}\left|\|\boldsymbol{W}^*\boldsymbol{Y}_N\|_4^4 - \mathbb{E}\|\boldsymbol{W}^*\boldsymbol{Y}_N\|_4^4\right| \geq \delta\right) \\
<&\exp\left[-\frac{pn^2\delta^2}{32R_2 + 8R_1 n\delta/3} + n^2\ln\left(\frac{12(L_f+\bar{L}_f)}{n\delta}\right) + \ln 2\right] + \mu(n,p) \\
<&\exp\left[-\frac{3p\delta^2}{C(1+\eta^2)^4 + 8n(\ln p)^4\delta} + n^2\ln\left(\frac{60n(\ln p)^4}{\delta}\right) + \ln 2\right] \\
&+ 2np\theta\exp\left(-\frac{(\ln p)^2}{4}\right) + 2np(1-\theta)\exp\left(-\frac{(\ln p)^2}{2}\right) < \frac{1}{p},
\end{aligned}
\tag{52}
$$

for a constant $C > 10^4$, when $p = \Omega\left((1+\eta^2)^4 n^2 \ln n/\delta^2\right)$. ■

## A.3    PROOF OF PROPOSITION 3.3

**Claim A.3 (Expectation of Objective with Outliers)** $\forall \theta \in (0,1)$, let $\boldsymbol{X}_o \in \mathbb{R}^{n\times p}$, $x_{i,j} \sim_{iid} BG(\theta)$. Let $\boldsymbol{D}_o \in \mathsf{O}(n;\mathbb{R})$ be an orthogonal matrix and assume $\boldsymbol{Y} = \boldsymbol{D}_o\boldsymbol{X}_o$. For any orthogonal matrix $\boldsymbol{W} \in \mathsf{O}(n;\mathbb{R})$ and any random Gaussian matrix $\boldsymbol{G}' \in \mathbb{R}^{n\times\tau p}$, $g'_{i,j} \sim_{iid} \mathcal{N}(0,1)$ independent of $\boldsymbol{X}_o$, let $\boldsymbol{Y}_O = [\boldsymbol{Y}, \boldsymbol{G}']$ denote the data with outliers $\boldsymbol{G}'$. Then the expectation of $\frac{1}{np}\|\boldsymbol{W}^*\boldsymbol{Y}_O\|_4^4$ satisfies:

$$\frac{1}{np}\mathbb{E}_{\boldsymbol{X}_o,\boldsymbol{G}'}\|\boldsymbol{W}^*\boldsymbol{Y}_O\|_4^4 = 3\theta(1-\theta)\frac{\|\boldsymbol{W}^*\boldsymbol{D}_o\|_4^4}{n} + 3\theta^2 + 3\tau. \tag{53}$$

**Proof** Notice that

$$\mathbb{E}_{\boldsymbol{X}_o,\boldsymbol{G}'}\|\boldsymbol{W}^*\boldsymbol{Y}_O\|_4^4 = \mathbb{E}_{\boldsymbol{X}_o}\|\boldsymbol{W}^*\boldsymbol{Y}\|_4^4 + \mathbb{E}_{\boldsymbol{G}'}\|\boldsymbol{W}^*\boldsymbol{G}'\|_4^4, \tag{54}$$

and

$$\mathbb{E}_{\boldsymbol{X}_o}\|\boldsymbol{W}^*\boldsymbol{Y}\|_4^4 = \mathbb{E}_{\boldsymbol{X}_o}\|\boldsymbol{W}^*\boldsymbol{D}_o\boldsymbol{X}_o\|_4^4 = 3p\theta(1-\theta)\|\boldsymbol{W}^*\boldsymbol{D}_o\|_4^4 + 3np\theta^2. \tag{55}$$

Moreover, the orthogonal rotation $(\boldsymbol{W}^*\boldsymbol{G}')$ of a standard Gaussian matrix $\boldsymbol{G}'$ is also a standard Gaussian matrix, therefore,

$$\mathbb{E}_{\boldsymbol{G}'}\|\boldsymbol{W}^*\boldsymbol{G}'\|_4^4 = 3\tau np. \tag{56}$$

Hence,

$$\frac{1}{np}\mathbb{E}_{\boldsymbol{X}_o,\boldsymbol{G}'}\|\boldsymbol{W}^*\boldsymbol{Y}_O\|_4^4 = 3\theta(1-\theta)\frac{\|\boldsymbol{W}^*\boldsymbol{D}_o\|_4^4}{n} + 3\theta^2 + 3\tau, \tag{57}$$

which completes the proof. ■

### A.4 PROOF OF THEOREM 3.4

**Claim A.4 (Concentration of Objective with Outliers)** $\forall \theta \in (0,1)$, let $\boldsymbol{X}_o \in \mathbb{R}^{n \times p}$, $x_{i,j} \sim_{iid}$ $BG(\theta)$. Let $\boldsymbol{D}_o \in \mathsf{O}(n; \mathbb{R})$ be an orthogonal matrix and assume $\boldsymbol{Y} = \boldsymbol{D}_o \boldsymbol{X}_o$. For any orthogonal matrix $\boldsymbol{W} \in \mathsf{O}(n; \mathbb{R})$ and any random Gaussian matrix $\boldsymbol{G}' \in \mathbb{R}^{n \times \tau p}$, $g'_{i,j} \sim_{iid} \mathcal{N}(0,1)$ independent of $\boldsymbol{X}_o$, let $\boldsymbol{Y}_O = [\boldsymbol{Y}, \boldsymbol{G}']$ denote the input with outlier $\boldsymbol{G}'$, then:

$$\mathbb{P}\left(\sup_{\boldsymbol{W} \in \mathsf{O}(n;\mathbb{R})} \frac{1}{np} \left| \|\boldsymbol{W}^*\boldsymbol{Y}_O\|_4^4 - \mathbb{E}\|\boldsymbol{W}^*\boldsymbol{Y}_O\|_4^4 \right| \geq \delta \right) < \frac{1}{p}, \tag{58}$$

*when* $p = \Omega(\tau^2 n^2 \ln n / \delta^2)$.

**Proof** Notice that

$$\mathbb{P}\left(\sup_{\boldsymbol{W} \in \mathsf{O}(n;\mathbb{R})} \frac{1}{np} \left| \|\boldsymbol{W}^*\boldsymbol{Y}_O\|_4^4 - \mathbb{E}\|\boldsymbol{W}^*\boldsymbol{Y}_O\|_4^4 \right| \geq \delta \right)$$

$$\leq \underbrace{\mathbb{P}\left(\sup_{\boldsymbol{W} \in \mathsf{O}(n;\mathbb{R})} \frac{1}{np} \left| \|\boldsymbol{W}^*\boldsymbol{Y}\|_4^4 - \mathbb{E}\|\boldsymbol{W}^*\boldsymbol{Y}\|_4^4 \right| \geq \frac{\delta}{2} \right)}_{\Gamma_1} \tag{59}$$

$$+ \underbrace{\mathbb{P}\left(\sup_{\boldsymbol{W} \in \mathsf{O}(n;\mathbb{R})} \frac{1}{np} \left| \|\boldsymbol{W}^*\boldsymbol{G}'\|_4^4 - \mathbb{E}\|\boldsymbol{W}^*\boldsymbol{G}'\|_4^4 \right| \geq \frac{\delta}{2} \right)}_{\Gamma_2}.$$

Apply Lemma B.5 (substitute $\delta$ with $\frac{\delta}{2}$) to $\Gamma_1$, we know that

$$\Gamma_1 = \mathbb{P}\left(\sup_{\boldsymbol{W} \in \mathsf{O}(n;\mathbb{R})} \frac{1}{np} \left| \|\boldsymbol{W}^*\boldsymbol{Y}\|_4^4 - \mathbb{E}\|\boldsymbol{W}^*\boldsymbol{Y}\|_4^4 \right| \geq \frac{\delta}{2} \right)$$

$$< \exp\left( -\frac{3p\delta^2}{c_1\theta + 16n(\ln p)^4\delta} + n^2 \ln\left(\frac{120np(\ln p)^4}{\delta}\right) \right) \tag{60}$$

$$+ \exp\left( -\frac{p\delta^2}{c_2\theta} + n^2 \ln\left(\frac{120np(\ln p)^4}{\delta}\right) \right) + 2np\theta \exp\left( -\frac{(\ln p)^2}{2} \right) < \frac{1}{2p},$$

for two constants $c_1 \geq 4 \times 10^4$, $c_2 > 13,440$, and the last inequality holds when $p = \Omega(\theta n^2 \ln n / \delta^2)$. Moreover, apply Lemma B.5 again (let $\theta = 1$, $\boldsymbol{Y} = \boldsymbol{G}' \in \mathbb{R}^{n \times \tau p}$ and substitute $\delta$ with $\frac{\delta}{2\tau}$), we have

$$\Gamma_2 = \mathbb{P}\left(\sup_{\boldsymbol{W} \in \mathsf{O}(n;\mathbb{R})} \frac{1}{np} \left| \|\boldsymbol{W}^*\boldsymbol{G}'\|_4^4 - \mathbb{E}\|\boldsymbol{W}^*\boldsymbol{G}'\|_4^4 \right| \geq \frac{\delta}{2} \right)$$

$$= \mathbb{P}\left(\sup_{\boldsymbol{W} \in \mathsf{O}(n;\mathbb{R})} \frac{1}{\tau np} \left| \|\boldsymbol{W}^*\boldsymbol{G}'\|_4^4 - \mathbb{E}\|\boldsymbol{W}^*\boldsymbol{G}'\|_4^4 \right| \geq \frac{\delta}{2\tau} \right)$$

$$< \exp\left( -\frac{3p(\delta/\tau)^2}{c_1 + 16n(\ln p)^4\delta/\tau} + n^2 \ln\left(\frac{120np(\ln p)^4}{\delta/\tau}\right) \right) \tag{61}$$

$$+ \exp\left( -\frac{p(\delta/\tau)^2}{c_2} + n^2 \ln\left(\frac{120np(\ln p)^4}{\delta/\tau}\right) \right) + 2np \exp\left( -\frac{(\ln p)^2}{2} \right) < \frac{1}{2p},$$

for two constants $c_1 \geq 4 \times 10^4$, $c_2 > 13,440$, and the last inequality holds when $p = \Omega(\tau^2 n^2 \ln n / \delta^2)$. Combine $\theta \leq 1$ and $\tau$ can be larger than 1, we conclude that

$$\mathbb{P}\left(\sup_{\boldsymbol{W} \in \mathsf{O}(n;\mathbb{R})} \frac{1}{np} \left| \|\boldsymbol{W}^*\boldsymbol{Y}_O\|_4^4 - \mathbb{E}\|\boldsymbol{W}^*\boldsymbol{Y}_O\|_4^4 \right| \geq \delta \right) < \Gamma_1 + \Gamma_2 < \frac{1}{p} \tag{62}$$

when $p = \Omega\left(\tau^2 n^2 \ln n / \delta^2\right)$. ∎

## A.5 PROOF OF PROPOSITION 3.5

**Claim A.5 (Expectation of Objective with Sparse Corruptions)** $\forall \theta \in (0,1)$, let $\boldsymbol{X}_o \in \mathbb{R}^{n \times p}$, $x_{i,j} \sim_{iid} BG(\theta)$. Let $\boldsymbol{D}_o \in \mathsf{O}(n; \mathbb{R})$ be an orthogonal matrix and assume $\boldsymbol{Y} = \boldsymbol{D}_o \boldsymbol{X}_o$. For any orthogonal matrix $\boldsymbol{W} \in \mathsf{O}(n; \mathbb{R})$ and any random Bernoulli matrix $\boldsymbol{B} \in \mathbb{R}^{n \times p}$, $b_{i,j} \sim_{iid} Ber(\beta)$ independent of $\boldsymbol{X}_o$, let $\boldsymbol{Y}_C = \boldsymbol{Y} + \sigma \boldsymbol{B} \circ \boldsymbol{S}$ denote the data with sparse corruptions, and $\boldsymbol{S} \in \mathbb{R}^{n \times p}$ is defined in equation 15. Then the expectation of $\frac{1}{np} \|\boldsymbol{W}^* \boldsymbol{Y}_C\|_4^4$ satisfies:

$$\frac{1}{np} \mathbb{E}_{\boldsymbol{X}_o, \boldsymbol{B}, \boldsymbol{S}} \|\boldsymbol{W}^* \boldsymbol{Y}_C\|_4^4 = 3\theta(1-\theta)\frac{\|\boldsymbol{W}^* \boldsymbol{D}_o\|_4^4}{n} + \sigma^4 \beta(1-3\beta)\frac{\|\boldsymbol{W}\|_4^4}{n} + 3\theta^2 + 6\sigma^2\theta\beta + 3\sigma^4\beta^2. \tag{63}$$

**Proof** Let $\boldsymbol{W}^* \boldsymbol{D}_o = \boldsymbol{M} \in \mathsf{O}(n; \mathbb{R})$, notice that

$$\|\boldsymbol{W}^* \boldsymbol{Y}_C\|_4^4 = \|\boldsymbol{M}\boldsymbol{X}_o + \sigma \boldsymbol{W}^*(\boldsymbol{B} \circ \boldsymbol{S})\|_4^4, \tag{64}$$

hence

$$\mathbb{E}_{\boldsymbol{X}_o, \boldsymbol{B}, \boldsymbol{S}} \|\boldsymbol{W}^* \boldsymbol{Y}_C\|_4^4 = \mathbb{E}_{\boldsymbol{X}_o, \boldsymbol{B}, \boldsymbol{S}} \sum_{j=1}^p \sum_{i=1}^n \left( \sum_{k=1}^n m_{i,k} x_{k,j} + \sigma \sum_{k=1}^n w_{k,i} b_{k,j} s_{k,j} \right)^4$$

$$= \sum_{j=1}^p \sum_{i=1}^n \mathbb{E}_{\boldsymbol{X}_o, \boldsymbol{B}, \boldsymbol{S}} \left[ \underbrace{\left( \sum_{k=1}^n m_{i,k} x_{k,j} \right)^4}_{\Gamma_1} + 6 \underbrace{\left( \sum_{k=1}^n m_{i,k} x_{k,j} \right)^2 \left( \sigma \sum_{k=1}^n w_{k,i} b_{k,j} s_{k,j} \right)^2}_{\Gamma_2} \right.$$

$$+ \underbrace{\left( \sigma \sum_{k=1}^n w_{k,i} b_{k,j} s_{k,j} \right)^4}_{\Gamma_3} + 4 \underbrace{\left( \sum_{k=1}^n m_{i,k} x_{k,j} \right)^3 \left( \sigma \sum_{k=1}^n w_{k,i} b_{k,j} s_{k,j} \right)}_{\Gamma_4} \tag{65}$$

$$+ 4 \underbrace{\left( \sum_{k=1}^n m_{i,k} x_{k,j} \right) \left( \sigma \sum_{k=1}^n w_{k,i} b_{k,j} s_{k,j} \right)^3}_{\Gamma_5} \Bigg].$$

Moreover,

- $\Gamma_1$ :

$$\mathbb{E}_{\boldsymbol{X}_o, \boldsymbol{B}, \boldsymbol{S}} \Gamma_1 = \mathbb{E}_{\boldsymbol{X}_o} \Gamma_1 = 3\theta \sum_{k=1}^n m_{i,k}^4 + 6\theta^2 \sum_{1 \le k_1 < k_2 \le n} m_{i,k_1}^2 m_{i,k_2}^2$$

$$= 3\theta(1-\theta) \sum_{k=1}^n m_{i,k}^4 + 3\theta^2, \tag{66}$$

- $\Gamma_2$ :

$$\mathbb{E}_{\boldsymbol{X}_o, \boldsymbol{B}, \boldsymbol{S}} \Gamma_2 = \left[ \mathbb{E}_{\boldsymbol{X}_o} \left( \sum_{k=1}^n m_{i,k} x_{k,j} \right)^2 \right] \left[ \mathbb{E}_{\boldsymbol{B}, \boldsymbol{S}} \left( \sigma \sum_{k=1}^n w_{k,i} b_{k,j} s_{k,j} \right)^2 \right]$$

$$= \theta \left( \sum_{k=1}^n m_{i,k}^2 \right) \sigma^2 \beta \left( \sum_{k=1}^n w_{k,i}^2 \right) = \sigma^2 \theta \beta, \tag{67}$$

- $\Gamma_3$ :

$$\mathbb{E}_{\boldsymbol{X}_o, \boldsymbol{B}, \boldsymbol{S}} \Gamma_3 = \mathbb{E}_{\boldsymbol{B}, \boldsymbol{S}} \Gamma_3 = \sigma^4 \beta \sum_{k=1}^n w_{k,i}^4 + 6\sigma^4 \beta^2 \sum_{1 \le k_1 < k_2 \le n} w_{k_1,i}^2 w_{k_2,i}^2$$

$$= \sigma^4 \beta(1-3\beta) \sum_{k=1}^n w_{k,i}^4 + 3\sigma^4 \beta^2, \tag{68}$$

- $\Gamma_4, \Gamma_5$ :

$$\mathbb{E}_{\boldsymbol{X}_o, \boldsymbol{B}, \boldsymbol{S}} \Gamma_4 = 0, \quad \mathbb{E}_{\boldsymbol{X}_o, \boldsymbol{B}, \boldsymbol{S}} \Gamma_5 = 0. \tag{69}$$

Substitute $\mathbb{E}\Gamma_1, \mathbb{E}\Gamma_2, \mathbb{E}\Gamma_3, \mathbb{E}\Gamma_4, \mathbb{E}\Gamma_5$ back to equation 65, yields

$$\frac{1}{np} \mathbb{E}_{\boldsymbol{X}_o, \boldsymbol{B}, \boldsymbol{S}} \|\boldsymbol{W}\boldsymbol{Y}_C\|_4^4 = 3\theta(1-\theta)\frac{\|\boldsymbol{W}^*\boldsymbol{D}_o\|_4^4}{n} + \sigma^4 \beta(1-3\beta)\frac{\|\boldsymbol{W}\|_4^4}{n} + 3\theta^2 + 6\sigma^2\theta\beta + 3\sigma^4\beta^2. \tag{70}$$

■

### A.6 Proof of Theorem 3.6

**Claim A.6 (Concentration of Objective with Sparse Corruptions)** $\forall \theta \in (0, 1)$, let $\boldsymbol{X}_o \in \mathbb{R}^{n \times p}$, $x_{i,j} \sim_{iid} BG(\theta)$. Let $\boldsymbol{D}_o \in \mathsf{O}(n; \mathbb{R})$ be an orthogonal matrix and assume $\boldsymbol{Y} = \boldsymbol{D}_o \boldsymbol{X}_o$. For any orthogonal matrix $\boldsymbol{W} \in \mathsf{O}(n; \mathbb{R})$ and any random Bernoulli matrix $\boldsymbol{B} \in \mathbb{R}^{n \times p}, b_{i,j} \sim_{iid} Ber(\beta)$ independent of $\boldsymbol{X}_o$, let $\boldsymbol{Y}_C = \boldsymbol{Y} + \sigma \boldsymbol{B} \circ \boldsymbol{S}$ denote the input with sparse corruptions, and $\boldsymbol{S} \in \mathbb{R}^{n \times p}$ is defined in equation 15, then:

$$\mathbb{P}\left(\sup_{\boldsymbol{W} \in \mathsf{O}(n; \mathbb{R})} \frac{1}{np} \left| \|\boldsymbol{W}^*\boldsymbol{Y}_C\|_4^4 - \mathbb{E}\|\boldsymbol{W}^*\boldsymbol{Y}_C\|_4^4 \right| \geq \delta \right) < \frac{1}{p}, \tag{71}$$

when $p = \Omega\left(\sigma^8 \beta n^2 \ln n/\delta^2\right)$.

**Proof** According to Proposition 3.5, we know that

$$\frac{1}{np} \mathbb{E}_{\boldsymbol{X}_o, \boldsymbol{B}, \boldsymbol{S}} \|\boldsymbol{W}^*\boldsymbol{Y}_C\|_4^4 = 3\theta(1-\theta)\frac{\|\boldsymbol{W}^*\boldsymbol{D}_o\|_4^4}{n} + \sigma^4 \beta(1-3\beta)\frac{\|\boldsymbol{W}\|_4^4}{n} + 3\theta^2 + 6\sigma^2\theta\beta + 3\sigma^4\beta^2. \tag{72}$$

Moreover, since $\boldsymbol{Y}_C = [\boldsymbol{y}_1 + \sigma\boldsymbol{b}_1 \circ \boldsymbol{s}_1, \boldsymbol{y}_2 + \sigma\boldsymbol{b}_2 \circ \boldsymbol{s}_2, \ldots, \boldsymbol{y}_p + \sigma\boldsymbol{b}_p \circ \boldsymbol{s}_p]$, whose columns are independent. Let

$$\boldsymbol{Z} = [\boldsymbol{z}_1, \boldsymbol{z}_2, \ldots, \boldsymbol{z}_p] = \boldsymbol{D}_o^* \boldsymbol{Y}_C = \boldsymbol{X}_o + \sigma\boldsymbol{C}, \tag{73}$$

where $\boldsymbol{C} = \boldsymbol{D}_o^*(\boldsymbol{B} \circ \boldsymbol{S})$. Note that the columns of $\boldsymbol{Z}$ are independent, we can define function $f_{\boldsymbol{z}}(\boldsymbol{W}) : \mathsf{O}(n; \mathbb{R}) \mapsto \mathbb{R}$ as

$$f_{\boldsymbol{z}}(\boldsymbol{W}) = \|\boldsymbol{W}^*\boldsymbol{z}\|_4^4. \tag{74}$$

Next, we will check assumption 1, 2, and 3 then apply Lemma B.4.

**Assumption 1:** $\mu(n, p)$. Since $\boldsymbol{Z} = \boldsymbol{X}_o + \sigma\boldsymbol{C}$, we have

$$\{\max_{i,j} |z_{i,j}| > B\} = \{\max_{i,j} |x_{i,j} + \sigma c_{i,j}| > B\} \subseteq \{\max_{i,j} |x_{i,j}| > B - \sigma \max_{i,j} |c_{i,j}|\}. \tag{75}$$

Moreover, since

$$\max_{i,j} |c_{i,j}| \leq \max_i \|\boldsymbol{d}_i\|_1 \leq \sqrt{n}, \tag{76}$$

we have

$$\{\max_{i,j} |z_{i,j}| > B\} \subseteq \{\max_{i,j} |x_{i,j}| > B - \sigma\sqrt{n}\}. \tag{77}$$

Thus we know that

$$\mathbb{P}\left(\max_{i,j} |z_{i,j}| > B\right) \leq \mathbb{P}\left(\max_{i,j} |x_{i,j}| > B - \sigma\sqrt{n}\right) \leq 2np\theta \exp\left(-\frac{(B - \sigma\sqrt{n})^2}{2}\right). \tag{78}$$

Specifically, we set $B = p^{\frac{1}{4}}$, which yields

$$\mu(n, p) = 2np\theta \exp\left(-\frac{\left(p^{\frac{1}{4}} - \sigma\sqrt{n}\right)^2}{2}\right). \tag{79}$$

**Assumption 2: Lipschitz Constant $L_f$.** By Proposition 3.5, $\forall \boldsymbol{W} \in \mathsf{O}(n; \mathbb{R})$, we know that

$$\mathbb{E} f_{\boldsymbol{z}}(\boldsymbol{W}) = 3\theta(1-\theta) \|\boldsymbol{W}^* \boldsymbol{D}_o\|_4^4 + \sigma^4 \beta(1-3\beta) \|\boldsymbol{W}\|_4^4 + 3n\theta^2 + 6n\sigma^2\theta\beta + 3n\sigma^4\beta^2. \quad (80)$$

Hence $\forall \boldsymbol{W}_1, \boldsymbol{W}_2 \in \mathsf{O}(n; \mathbb{R})$, we have

$$\begin{aligned} &\left| \mathbb{E} \|\boldsymbol{W}_1^* \boldsymbol{z}\|_4^4 - \mathbb{E} \|\boldsymbol{W}_w^* \boldsymbol{z}\|_4^4 \right| \\ \leq & 3\theta(1-\theta) \left| \|\boldsymbol{W}_1^* \boldsymbol{D}_o\|_4^4 - \|\boldsymbol{W}_1^* \boldsymbol{D}_o\|_4^4 \right| + \sigma^4 \beta(1-3\beta) \left| \|\boldsymbol{W}_1\|_4^4 - \|\boldsymbol{W}_2\|_4^4 \right|, \end{aligned} \quad (81)$$

from the derivation of equation 34, we have

$$\left| \|\boldsymbol{W}_1^* \boldsymbol{D}_o\|_4^4 - \|\boldsymbol{W}_2^* \boldsymbol{D}_o\|_4^4 \right| \leq 4n^2 \|\boldsymbol{W}_1 - \boldsymbol{W}_2\|_2, \quad (82)$$

and therefore, we have

$$\left| \mathbb{E} \|\boldsymbol{W}_1^* \boldsymbol{z}\|_4^4 - \mathbb{E} \|\boldsymbol{W}_w^* \boldsymbol{z}\|_4^4 \right| \leq \left[ 12n^2 \theta(1-\theta) + 4n^2 \sigma^4 \beta(1-3\beta) \right] \|\boldsymbol{W}_1 - \boldsymbol{W}_2\|_2, \quad (83)$$

which yields

$$L_f = 12n^2 \theta(1-\theta) + 4n^2 \sigma^4 \beta(1-3\beta). \quad (84)$$

**Assumption 2: Lipschitz Constant $\bar{L}_f$.** Since $f_{\bar{\boldsymbol{z}}}(\boldsymbol{W}) = \|\boldsymbol{W}^* \bar{\boldsymbol{z}}\|_4^4$, apply the same derivation as equation 37, we have

$$\left| \|\boldsymbol{W}_1^* \bar{\boldsymbol{z}}\|_4^4 - \|\boldsymbol{W}_2^* \bar{\boldsymbol{z}}\|_4^4 \right| \leq 4n^2 B^4 \|\boldsymbol{W}_1 - \boldsymbol{W}_2\|_2, \quad (85)$$

which yields

$$\bar{L}_f = 4n^2 B^4. \quad (86)$$

**Assumption 3: Upper Bound $R_1$.** Notice that $\forall \boldsymbol{W} \in \mathsf{O}(n; \mathbb{R})$,

$$f_{\bar{\boldsymbol{z}}}(\boldsymbol{W}) = \|\boldsymbol{W}^* \bar{\boldsymbol{z}}\|_4^4 = \|\boldsymbol{W}^* \bar{\boldsymbol{z}}\|_4^4 = \|\boldsymbol{W}^* \bar{\boldsymbol{z}}\|_2^4 \left\| \frac{\boldsymbol{W}^* \bar{\boldsymbol{z}}}{\|\boldsymbol{W}^* \bar{\boldsymbol{z}}\|_2} \right\|_4^4 \leq \|\boldsymbol{W}^* \bar{\boldsymbol{z}}\|_2^4 = \|\bar{\boldsymbol{z}}\|_2^4 \leq n^2 B^4 = R_1. \quad (87)$$

**Assumption 3: Upper Bound $R_2$.** Assume the support for $\boldsymbol{x}$ is $\mathcal{S}$, so we know that $\forall i \in [n]$,

$$z_i = \begin{cases} v_i + c_i, & \text{if } i \in \mathcal{S}, \\ c_i, & \text{otherwise,} \end{cases} \quad (88)$$

where $\boldsymbol{v} \sim \mathcal{N}(\boldsymbol{0}, \boldsymbol{I})$ is a Gaussian vector and $\boldsymbol{c} = \boldsymbol{D}_o^*(\boldsymbol{b} \circ \boldsymbol{s})$ is the sparse corruption vector after orthogonal rotation $\boldsymbol{D}_o^*$. Let $\boldsymbol{P}_{\mathcal{S}} : \mathbb{R}^n \mapsto \mathbb{R}^n$ be a projection onto the support $\mathcal{S}$, that is, $\forall \boldsymbol{q} \in \mathbb{R}^n$:

$$(\boldsymbol{P}_{\mathcal{S}} \boldsymbol{q})_i = \begin{cases} q_i, & \text{if } i \in \mathcal{S}, \\ 0, & \text{otherwise.} \end{cases} \quad (89)$$

Let $\boldsymbol{M} = \boldsymbol{D}_o \boldsymbol{W}$, and $\boldsymbol{h} = \boldsymbol{b} \circ \boldsymbol{s}$, we have

$$\begin{aligned} \mathbb{E} f_{\boldsymbol{z}}^2(\boldsymbol{W}) &= \mathbb{E} \left( \|\boldsymbol{W}^* \boldsymbol{z}\|_4^8 \right) \\ &= \mathbb{E} \sum_{i=1}^n \sum_{j=1}^n \left[ (\langle \boldsymbol{w}_i, \boldsymbol{x} \rangle + \sigma \langle \boldsymbol{m}_i, \boldsymbol{h} \rangle)^4 (\langle \boldsymbol{w}_j, \boldsymbol{x} \rangle + \sigma \langle \boldsymbol{m}_j, \boldsymbol{h} \rangle)^4 \right] \\ &\leq \sum_{i=1}^n \sum_{j=1}^n \left[ \mathbb{E} (\langle \boldsymbol{w}_i, \boldsymbol{x} \rangle + \sigma \langle \boldsymbol{m}_i, \boldsymbol{h} \rangle)^8 \, \mathbb{E} (\langle \boldsymbol{w}_j, \boldsymbol{x} \rangle + \sigma \langle \boldsymbol{m}_j, \boldsymbol{h} \rangle)^8 \right]^{\frac{1}{2}} \\ &= \sum_{i=1}^n \sum_{j=1}^n \left[ \mathbb{E} (\langle \boldsymbol{P}_{\mathcal{S}} \boldsymbol{w}_i, \boldsymbol{v} \rangle + \sigma \langle \boldsymbol{m}_i, \boldsymbol{h} \rangle)^8 \, \mathbb{E} (\langle \boldsymbol{P}_{\mathcal{S}} \boldsymbol{w}_j, \boldsymbol{v} \rangle + \sigma \langle \boldsymbol{m}_j, \boldsymbol{h} \rangle)^8 \right]^{\frac{1}{2}}. \end{aligned} \quad (90)$$

Moreover, we have:

$$
\begin{aligned}
&\mathbb{E}\left(\langle \boldsymbol{P}_{\mathcal{S}}\boldsymbol{w}_i, \boldsymbol{v}\rangle + \sigma \langle \boldsymbol{m}_i, \boldsymbol{h}\rangle\right)^8 \\
=&\mathbb{E}\left(\langle \boldsymbol{P}_{\mathcal{S}}\boldsymbol{w}_i, \boldsymbol{v}\rangle^8\right) + 28\sigma^2 \mathbb{E}\left(\langle \boldsymbol{P}_{\mathcal{S}}\boldsymbol{w}_i, \boldsymbol{v}\rangle^6 \langle \boldsymbol{m}_i, \boldsymbol{h}\rangle^2\right) + 70\sigma^4 \mathbb{E}\left(\langle \boldsymbol{P}_{\mathcal{S}}\boldsymbol{w}_i, \boldsymbol{v}\rangle^4 \langle \boldsymbol{m}_i, \boldsymbol{h}\rangle^4\right) \\
&+ 28\sigma^6 \mathbb{E}\left(\langle \boldsymbol{P}_{\mathcal{S}}\boldsymbol{w}_i, \boldsymbol{v}\rangle^6 \langle \boldsymbol{m}_i, \boldsymbol{h}\rangle^2\right) + \sigma^8 \mathbb{E}\left(\langle \boldsymbol{m}_i, \boldsymbol{h}\rangle^8\right).
\end{aligned}
\tag{91}
$$

We now provide upper bound for $\mathbb{E}\left(\langle \boldsymbol{P}_{\mathcal{S}}\boldsymbol{w}_i, \boldsymbol{v}\rangle^r\right)$ and $\mathbb{E}\left(\langle \boldsymbol{m}_i, \boldsymbol{h}\rangle^r\right)$ for $r = 2, 4, 6, 8$ respectively. Since $\mathbb{E}\boldsymbol{h} = \boldsymbol{0}$ and $\boldsymbol{h} = \boldsymbol{b} \circ \boldsymbol{s}$. Let $\mathcal{S}'$ denote the support of $\boldsymbol{h}$, we have

$$
\begin{aligned}
\mathbb{E}\left(\langle \boldsymbol{m}_i, \boldsymbol{h}\rangle^8\right) &= \mathbb{E}\prod_{l=1}^{8}\left(\sum_{k_1=1}^{n} m_{k_l,i} h_i\right) \\
&= \sum_{k_1,k_2,k_3,k_4} \mathbb{E}_{\mathcal{S}'}\left(m_{k_1,i}^2 \mathbb{1}_{k_1\in\mathcal{S}'} m_{k_2,i}^2 \mathbb{1}_{k_2\in\mathcal{S}'} m_{k_3,i}^2 \mathbb{1}_{k_3\in\mathcal{S}'} m_{k_4,i}^2 \mathbb{1}_{k_4\in\mathcal{S}'}\right).
\end{aligned}
\tag{92}
$$

Now we discuss these cases separately:

- With probability $c_1\beta^4(c_1 \leq 1)$, all $k_1, k_2, k_3, k_4$ are in $\mathcal{S}$, in this case, we have

$$
\begin{aligned}
&\sum_{k_1,k_2,k_3,k_4} \mathbb{E}_{\mathcal{S}'}\left(m_{k_1,i}^2 \mathbb{1}_{k_1\in\mathcal{S}'} m_{k_2,i}^2 \mathbb{1}_{k_2\in\mathcal{S}'} m_{k_3,i}^2 \mathbb{1}_{k_3\in\mathcal{S}'} m_{k_4,i}^2 \mathbb{1}_{k_4\in\mathcal{S}'}\right) \\
&= \sum_{k_1,k_2,k_3,k_4}\left(m_{k_1,i}^2 m_{k_2,i}^2 m_{k_3,i}^2 m_{k_4,i}^2\right) = 1.
\end{aligned}
\tag{93}
$$

- With probability $c_2\beta^3(c_2 \leq 1)$, only three among $k_1, k_2, k_3, k_4$ are in $\mathcal{S}'$, in this case, we have

$$
\begin{aligned}
&\sum_{k_1,k_2,k_3,k_4} \mathbb{E}_{\mathcal{S}'}\left(m_{k_1,i}^2 \mathbb{1}_{k_1\in\mathcal{S}'} m_{k_2,i}^2 \mathbb{1}_{k_2\in\mathcal{S}'} m_{k_3,i}^2 \mathbb{1}_{k_3\in\mathcal{S}'} m_{k_4,i}^2 \mathbb{1}_{k_4\in\mathcal{S}'}\right) \\
&= \sum_{k_1,k_2,k_3}\left(m_{k_1,i}^2 m_{k_2,i}^2 m_{k_3,i}^4\right) = \|\boldsymbol{m}_i\|_4^4 \leq 1.
\end{aligned}
\tag{94}
$$

- With probability $c_3\beta^2(c_3 \leq 1)$, only two among $k_1, k_2, k_3, k_4$ are in $\mathcal{S}'$, in this case, we have

$$
\begin{aligned}
&\sum_{k_1,k_2,k_3,k_4} \mathbb{E}_{\mathcal{S}'}\left(m_{k_1,i}^2 \mathbb{1}_{k_1\in\mathcal{S}'} m_{k_2,i}^2 \mathbb{1}_{k_2\in\mathcal{S}'} m_{k_3,i}^2 \mathbb{1}_{k_3\in\mathcal{S}'} m_{k_4,i}^2 \mathbb{1}_{k_4\in\mathcal{S}'}\right) \\
&= \sum_{k_1,k_2}\left(m_{k_1,i}^4 m_{k_2,i}^4\right) + \sum_{k_1,k_2}\left(m_{k_1,i}^2 m_{k_2,i}^6\right) \leq \|\boldsymbol{m}_i\|_4^4 + \|\boldsymbol{m}_i\|_6^6 \leq 2.
\end{aligned}
\tag{95}
$$

- With probability $c_4\beta(c_4 \leq 1)$, only one among $k_1, k_2, k_3, k_4$ is in $\mathcal{S}'$, in this case, we have

$$
\begin{aligned}
&\sum_{k_1,k_2,k_3,k_4} \mathbb{E}_{\mathcal{S}'}\left(m_{k_1,i}^2 \mathbb{1}_{k_1\in\mathcal{S}'} m_{k_2,i}^2 \mathbb{1}_{k_2\in\mathcal{S}'} m_{k_3,i}^2 \mathbb{1}_{k_3\in\mathcal{S}'} m_{k_4,i}^2 \mathbb{1}_{k_4\in\mathcal{S}'}\right) \\
&= \sum_{k_1} m_{k_1,i}^8 = \|\boldsymbol{m}_i\|_8^8 \leq 1.
\end{aligned}
\tag{96}
$$

Hence, combine equation 93, equation 94, equation 95, and equation 96, we know that

$$
\mathbb{E}\left(\langle \boldsymbol{m}_i, \boldsymbol{h}\rangle^8\right) \leq c\beta,
\tag{97}
$$

for a constant $c > 1$. Similarly, one can conclude that

$$
\mathbb{E}\left(\langle \boldsymbol{m}_i, \boldsymbol{h}\rangle^r\right) \leq c\beta, \quad \forall r = 2, 4, 6.
\tag{98}
$$

Moreover, since $\boldsymbol{v} \sim \mathcal{N}(\boldsymbol{0}, \boldsymbol{I})$, we know that

$$
\mathbb{E}\left(\langle \boldsymbol{P}_{\mathcal{S}}\boldsymbol{w}_i, \boldsymbol{v}\rangle^r\right) = \|\boldsymbol{P}_{\mathcal{S}}\boldsymbol{w}_i\|_2^r, \quad \forall r = 2, 4, 6, 8.
\tag{99}
$$

Hence, with the similar technique applied above, we have

$$
\begin{aligned}
\mathbb{E}\left(\langle \boldsymbol{P}_{\mathcal{S}}\boldsymbol{w}_i, \boldsymbol{v}\rangle^8\right) &= 105\mathbb{E}\left\|\boldsymbol{P}_{\mathcal{S}}\boldsymbol{w}_i\right\|_2^8 \\
&=105\sum_{k_1,k_2,k_3,k_4}\mathbb{E}_{\mathcal{S}'}\left(w_{k_1,i}^2\mathbb{1}_{k_1\in\mathcal{S}'}w_{k_2,i}^2\mathbb{1}_{k_2\in\mathcal{S}'}w_{k_3,i}^2\mathbb{1}_{k_3\in\mathcal{S}'}w_{k_4,i}^2\mathbb{1}_{k_4\in\mathcal{S}'}\right) \le 105c\theta,
\end{aligned}
\tag{100}
$$

for a constant $c > 1$. Similarly, we have

$$
\mathbb{E}\left(\langle \boldsymbol{P}_{\mathcal{S}}\boldsymbol{w}_i, \boldsymbol{v}\rangle^r\right) = \left\|\boldsymbol{P}_{\mathcal{S}}\boldsymbol{w}_i\right\|_2^r \le c(r-1)!!\theta, \quad \forall r = 2, 4, 6,
\tag{101}
$$

for a constant $c > 1$. Therefore, combine equation 91, we have

$$
\begin{aligned}
&\mathbb{E}\left(\langle \boldsymbol{P}_{\mathcal{S}}\boldsymbol{w}_i, \boldsymbol{v}\rangle + \sigma\langle \boldsymbol{m}_i, \boldsymbol{h}\rangle\right)^8 \\
=&\mathbb{E}\left(\langle \boldsymbol{P}_{\mathcal{S}}\boldsymbol{w}_i, \boldsymbol{v}\rangle^8\right) + 28\sigma^2\mathbb{E}\left(\langle \boldsymbol{P}_{\mathcal{S}}\boldsymbol{w}_i, \boldsymbol{v}\rangle^6\langle \boldsymbol{m}_i, \boldsymbol{h}\rangle^2\right) + 70\sigma^4\mathbb{E}\left(\langle \boldsymbol{P}_{\mathcal{S}}\boldsymbol{w}_i, \boldsymbol{v}\rangle^4\langle \boldsymbol{m}_i, \boldsymbol{h}\rangle^4\right) \\
&+ 28\sigma^6\mathbb{E}\left(\langle \boldsymbol{P}_{\mathcal{S}}\boldsymbol{w}_i, \boldsymbol{v}\rangle^6\langle \boldsymbol{m}_i, \boldsymbol{h}\rangle^2\right) + \sigma^8\mathbb{E}\left(\langle \boldsymbol{m}_i, \boldsymbol{h}\rangle^8\right) \\
\le& c\theta + 28c\sigma^2\theta\beta + 70c\sigma^4\theta\beta + 28c\sigma^6\theta\beta + c\sigma^8\beta \le c\sigma^8\beta,
\end{aligned}
\tag{102}
$$

for a constant $c > 1$. Hence, combine equation 90, we have

$$
\begin{aligned}
\mathbb{E}f_{\boldsymbol{z}}^2(\boldsymbol{W}) &= \mathbb{E}\left(\|\boldsymbol{W}^*\boldsymbol{z}\|_4^8\right) \\
&=\sum_{i=1}^n\sum_{j=1}^n\left[\mathbb{E}\left(\langle \boldsymbol{P}_{\mathcal{S}}\boldsymbol{w}_i, \boldsymbol{v}\rangle + \sigma\langle \boldsymbol{m}_i, \boldsymbol{h}\rangle\right)^8\mathbb{E}\left(\langle \boldsymbol{P}_{\mathcal{S}}\boldsymbol{w}_j, \boldsymbol{v}\rangle + \sigma\langle \boldsymbol{m}_j, \boldsymbol{h}\rangle\right)^8\right]^{\frac{1}{2}} \le c\sigma^8\beta n^2,
\end{aligned}
\tag{103}
$$

for a constant $c > 1$. Therefore, we can conclude that

$$
R_2 = c\sigma^8\beta n^2,
\tag{104}
$$

for a constant $c > 1$.

**Applying Lemma B.4 for Concentration.**     Now we apply Lemma B.4 with

1.
$$
B = p^{\frac{1}{4}}, \quad \mu(n,p) = 2np\theta\exp\left(-\frac{\left(p^{\frac{1}{4}}-\sigma\sqrt{n}\right)^2}{2}\right),
\tag{105}
$$

2.
$$
L_f = 12n^2\theta(1-\theta) + 4n^2\sigma^4\beta(1-3\beta), \quad \bar{L}_f = 4n^2p,
\tag{106}
$$

3.
$$
R_1 = n^2p, \quad R_2 = c\sigma^8\beta n^2,
\tag{107}
$$

for a constant $c > 1$, we have

$$
\begin{aligned}
&\mathbb{P}\left(\sup_{\boldsymbol{W}\in\mathsf{O}(n;\mathbb{R})}\frac{1}{np}\left|\sum_{j=1}^p\left[f_{\boldsymbol{z}_j}(\boldsymbol{W}) - \mathbb{E}f_{\boldsymbol{z}_j}(\boldsymbol{W})\right]\right| \ge \delta\right) \\
=&\mathbb{P}\left(\sup_{\boldsymbol{W}\in\mathsf{O}(n;\mathbb{R})}\frac{1}{np}\left|\|\boldsymbol{W}^*\boldsymbol{Y}_C\|_4^4 - \mathbb{E}\|\boldsymbol{W}^*\boldsymbol{Y}_C\|_4^4\right| \ge \delta\right) \\
<&\exp\left[-\frac{pn^2\delta^2}{32R_2 + 8R_1n\delta/3} + n^2\ln\left(\frac{12(L_f+\bar{L}_f)}{n\delta}\right) + \ln 2\right] + \mu(n,p) \\
<&\exp\left[-\frac{3p\delta^2}{C\sigma^8\beta + 8np\delta} + n^2\ln\left(\frac{60np}{\delta}\right) + \ln 2\right] \\
&+ 2np\theta\exp\left(-\frac{\left(p^{\frac{1}{4}}-\sigma\sqrt{n}\right)^2}{2}\right) < \frac{1}{p},
\end{aligned}
\tag{108}
$$

for a constant $C > 96$, when $p = \Omega\left(\sigma^8\beta n^2\ln n/\delta^2\right)$. $\blacksquare$

# B    RELATED LEMMAS AND INEQUALITIES

**Lemma B.1 (Two-sided Bernstein's Inequality)** *Given $p$ random variables $x_1, x_2, \ldots x_p$, if $\forall i \in [p], |x_i| \leq b$ almost surely, then*

$$\mathbb{P}\left( \frac{1}{p} \left| \sum_{i=1}^{p} [x_i - \mathbb{E}[x_i]] \right| \geq t \right) \leq 2 \exp\left( -\frac{pt^2}{\frac{2}{p} \sum_{i=1}^{p} \mathbb{E}[x_i^2] + 2bt/3} \right). \tag{109}$$

**Proof**  See Proposition 2.14 in Wainwright (2019), one can easily generalize it to two-sided case. ∎

**Lemma B.2 (Entry-wise Truncation of a Bernoulli Gaussian Matrix)** *Let $X \in \mathbb{R}^{n \times p}$, where $x_{i,j} \sim_{iid} BG(\theta)$ and let $\|\cdot\|_\infty$ denote the maximum element (in absolute value) of a matrix, then*

$$\mathbb{P}\left( \max_{i,j} |x_{i,j}| \geq t \right) \leq 2np\theta \exp\left( -\frac{t^2}{2} \right). \tag{110}$$

**Proof**  A Bernoulli Gaussian variable $x_{i,j}, \forall i \in [n], j \in [p]$ satisfies $x_{i,j} = b_{i,j} \cdot g_{i,j}$, where $b_{i,j} \sim_{iid} \text{Ber}(\theta)$, $g_{i,j} \sim_{iid} \mathcal{N}(0,1)$ and therefore

$$\mathbb{P}\left( |x_{i,j}| \geq t \right) = \theta \cdot \mathbb{P}\left( |g_{i,j}| \geq t \right) \leq 2\theta \exp\left( -\frac{t^2}{2} \right). \tag{111}$$

By union bound, we have:

$$\mathbb{P}\left( \max_{i,j} |x_{i,j}| \geq t \right) \leq \sum_{i=1}^{n} \sum_{j=1}^{p} \mathbb{P}\left( |x_{i,j}| \geq t \right) \leq 2np\theta \exp\left( -\frac{t^2}{2} \right). \tag{112}$$

∎

**Lemma B.3 ($\epsilon-$Net Covering of Stiefel Manifolds)** [11] *There is a covering $\epsilon-$net $\mathcal{S}_\epsilon$ for Stiefel manifold $\mathcal{M} = \{W \in \mathbb{R}^{n \times r} | W^* W = I\}, (n \geq r)$ in operator norm*

$$\forall W \in \mathcal{M}, \ \exists W' \in \mathcal{S}_\epsilon \quad \text{subject to} \quad \|W - W'\|_2 \leq \epsilon, \tag{113}$$

*of size $|\mathcal{S}_\epsilon| \leq \left( \frac{6}{\epsilon} \right)^{nr}$.*

**Proof**  See Lemma D.4 in Zhai et al. (2019b). ∎

**Lemma B.4 (Uniform Concentration bound over $\mathsf{O}(n; \mathbb{R})$)** *Let $Z \in \mathbb{R}^{n \times p}$ be a random matrix whose columns $z_1, z_2, \ldots, z_p$ are i.i.d. drawn from a distribution $\mathcal{P}$. $\forall z \sim \mathcal{P}$, let $f_z(\cdot) : \mathsf{O}(n; \mathbb{R}) \mapsto \mathbb{R}$ denote a function that maps $\mathsf{O}(n; \mathbb{R})$ to $\mathbb{R}$. $\forall B > 0$, let $\bar{Z}$ denote the truncation of $Z$:*

$$\bar{z}_{i,j} = \begin{cases} z_{i,j} & \text{if} \quad |z_{i,j}| \leq B, \\ 0 & \text{otherwise.} \end{cases} \tag{114}$$

*Assume that:*

1. *$\mathbb{P}\left( \max_{i,j} |z_{i,j}| > B \right) < \mu(n,p)$, where $\mu(n,p) \to 0$ as $p$ increase, $B$ depends on $p$.*

2. *$\mathbb{E}f_z(\cdot)$ is $L_f$-Lipschitz and $f_{\bar{z}}(\cdot)$ is $\bar{L}_f$-Lipschitz, that is, $\forall W_1, W_2 \in \mathsf{O}(n; \mathbb{R})$:*

$$\begin{aligned} |\mathbb{E}f_z(W_1) - \mathbb{E}f_z(W_2)| &\leq L_f \|W_1 - W_2\|_2, \\ |f_{\bar{z}}(W_1) - f_{\bar{z}}(W_2)| &\leq \bar{L}_f \|W_1 - W_2\|_2. \end{aligned} \tag{115}$$

3. *$\forall W \in \mathsf{O}(n; \mathbb{R})$, $f_{\bar{z}}(W) \leq R_1, \mathbb{E}[f_z^2(W)] \leq R_2$.*

---

[11] A similar result can be found in Lemma 4.5 of Recht et al. (2010).

*Then:*

$$\mathbb{P}\left(\sup_{\boldsymbol{W}\in\mathsf{O}(n;\mathbb{R})} \frac{1}{np}\left|\sum_{j=1}^{p}\left[f_{\boldsymbol{z}_j}(\boldsymbol{W}) - \mathbb{E}f_{\boldsymbol{z}_j}(\boldsymbol{W})\right]\right| \geq \delta\right)$$
$$< \exp\left[-\frac{pn^2\delta^2}{32R_2 + 8R_1 n\delta/3} + n^2\ln\left(\frac{12(L_f + \bar{L}_f)}{n\delta}\right) + \ln 2\right] + \mu(n,p), \tag{116}$$

*when $p > \rho$, where $\rho$ depends on $n$.*

**Proof** By assumption 1, we have

$$\mathbb{P}\left(\sup_{\boldsymbol{W}\in\mathsf{O}(n;\mathbb{R})} \frac{1}{np}\left|\sum_{j=1}^{p}\left[f_{\boldsymbol{z}_j}(\boldsymbol{W}) - \mathbb{E}f_{\boldsymbol{z}_j}(\boldsymbol{W})\right]\right| \geq \delta\right)$$
$$\leq \mathbb{P}(\boldsymbol{Z} \neq \bar{\boldsymbol{Z}}) + \mathbb{P}\left(\sup_{\boldsymbol{W}\in\mathsf{O}(n;\mathbb{R})}\left|\frac{1}{p}\sum_{j=1}^{p}f_{\bar{\boldsymbol{z}}_j}(\boldsymbol{W}) - \mathbb{E}f_{\boldsymbol{z}}(\boldsymbol{W})\right| \geq n\delta\right) \tag{117}$$
$$\leq \mu(n,p) + \mathbb{P}\left(\sup_{\boldsymbol{W}\in\mathsf{O}(n;\mathbb{R})}\left|\frac{1}{p}\sum_{j=1}^{p}f_{\bar{\boldsymbol{z}}_j}(\boldsymbol{W}) - \mathbb{E}f_{\boldsymbol{z}}(\boldsymbol{W})\right| \geq n\delta\right).$$

**Uniform bound over $\mathsf{O}(n;\mathbb{R})$.** $\forall \epsilon > 0$, lemma B.3 shows there exists an $\epsilon$-nets $\mathcal{S}_\epsilon$ that covers $\mathsf{O}(n;\mathbb{R})$:

$$\mathcal{S}_\epsilon = \{\boldsymbol{W}_1, \boldsymbol{W}_2, \ldots, \boldsymbol{W}_{|\mathcal{S}_\epsilon|}\}, \quad \mathsf{O}(n;\mathbb{R}) \subset \bigcup_{l=1}^{|\mathcal{S}_\epsilon|}\mathbb{B}(\boldsymbol{W}_l, \epsilon), \tag{118}$$

and $|\mathcal{S}_\epsilon|$ satisfies $|\mathcal{S}_\epsilon| \leq (6/\epsilon)^{n^2}$. Together with Lipschitz assumption 2, we know that

$$\sup_{\boldsymbol{W}\in\mathbb{B}(\boldsymbol{W}_l,\epsilon)}\left|\frac{1}{p}\sum_{j=1}^{p}f_{\bar{\boldsymbol{z}}_j}(\boldsymbol{W}) - \mathbb{E}f_{\boldsymbol{z}}(\boldsymbol{W})\right|$$
$$\leq \sup_{\boldsymbol{W}\in\mathbb{B}(\boldsymbol{W}_l,\epsilon)}\left|\frac{1}{p}\sum_{j=1}^{p}f_{\bar{\boldsymbol{z}}_j}(\boldsymbol{W}_l) - \mathbb{E}f_{\boldsymbol{z}}(\boldsymbol{W}_l)\right| + \sup_{\boldsymbol{W}\in\mathbb{B}(\boldsymbol{W}_l,\epsilon)}|\mathbb{E}f_{\boldsymbol{z}}(\boldsymbol{W}_l) - \mathbb{E}f_{\boldsymbol{z}}(\boldsymbol{W})|$$
$$+ \sup_{\boldsymbol{W}\in\mathbb{B}(\boldsymbol{W},\epsilon)}\left|\frac{1}{p}\sum_{j=1}^{p}f_{\bar{\boldsymbol{z}}_j}(\boldsymbol{W}_l) - \frac{1}{p}\sum_{j=1}^{p}f_{\bar{\boldsymbol{z}}_j}(\boldsymbol{W})\right| \tag{119}$$
$$\leq \sup_{\boldsymbol{W}\in\mathbb{B}(\boldsymbol{W}_l,\epsilon)}\left|\frac{1}{p}\sum_{j=1}^{p}f_{\bar{\boldsymbol{z}}_j}(\boldsymbol{W}_l) - \mathbb{E}f_{\boldsymbol{z}}(\boldsymbol{W}_l)\right| + (L_f + \bar{L}_f)\epsilon.$$

Hence, let

$$\epsilon = \frac{n\delta}{2(L_f + \bar{L}_f)}, \tag{120}$$

we have

$$
\mathbb{P}\left(\sup_{\boldsymbol{W}\in\mathsf{O}(n;\mathbb{R})}\left|\frac{1}{p}\sum_{j=1}^{p}f_{\bar{\boldsymbol{z}}_j}(\boldsymbol{W})-\mathbb{E}f_{\boldsymbol{z}}(\boldsymbol{W})\right|\geq n\delta\right)
$$

$$
\leq\sum_{l=1}^{|\mathcal{S}_\epsilon|}\mathbb{P}\left(\sup_{\boldsymbol{W}\in\mathbb{B}(\boldsymbol{W}_l;\epsilon)}\left|\frac{1}{p}\sum_{j=1}^{p}f_{\bar{\boldsymbol{z}}_j}(\boldsymbol{W})-\mathbb{E}f_{\boldsymbol{z}}(\boldsymbol{W})\right|\geq n\delta\right)
$$

$$
<\left(\frac{6}{\epsilon}\right)^{n^2}\mathbb{P}\left(\sup_{\boldsymbol{W}\in\mathbb{B}(\boldsymbol{W}_l;\epsilon)}\left|\frac{1}{p}\sum_{j=1}^{p}f_{\bar{\boldsymbol{z}}_j}(\boldsymbol{W}_l)-\mathbb{E}f_{\boldsymbol{z}}(\boldsymbol{W}_l)\right|\geq n\delta-(L_f+\bar{L}_f)\epsilon\right) \quad (121)
$$

$$
=\exp\left[n^2\ln\left(\frac{12(L_f+\bar{L}_f)}{n\delta}\right)\right]\mathbb{P}\left(\sup_{\boldsymbol{W}\in\mathbb{B}(\boldsymbol{W}_l;\epsilon)}\left|\frac{1}{p}\sum_{j=1}^{p}f_{\bar{\boldsymbol{z}}_j}(\boldsymbol{W}_l)-\mathbb{E}f_{\boldsymbol{z}}(\boldsymbol{W}_l)\right|\geq\frac{n\delta}{2}\right).
$$

**Tail bound within each $\mathbb{B}(\boldsymbol{W}_l,\epsilon)$.** Then, $\forall l\in[\mathcal{S}_\epsilon]$, we apply point-wise control to a given point $\boldsymbol{W}_l\in\mathcal{S}$. Later we will provide a uniform concentration bound over $\mathsf{O}(n;\mathbb{R})$. By triangle inequality, we have

$$
\left|\frac{1}{p}\sum_{j=1}^{p}f_{\bar{\boldsymbol{z}}_j}(\boldsymbol{W}_l)-\mathbb{E}f_{\boldsymbol{z}_j}(\boldsymbol{W}_l)\right|\leq\left|\frac{1}{p}\sum_{j=1}^{p}f_{\bar{\boldsymbol{z}}_j}(\boldsymbol{W}_l)-\mathbb{E}f_{\bar{\boldsymbol{z}}}(\boldsymbol{W}_l)\right|+\left|\mathbb{E}f_{\bar{\boldsymbol{z}}_j}(\boldsymbol{W}_l)-\mathbb{E}f_{\boldsymbol{z}}(\boldsymbol{W}_l)\right|.
$$
$$
(122)
$$

$\forall\boldsymbol{W}\in\mathbb{B}(\boldsymbol{W}_l,\epsilon)$, by assumption 1 and 3, we have

$$
\left|\mathbb{E}f_{\bar{\boldsymbol{z}}_j}(\boldsymbol{W})-\mathbb{E}f_{\boldsymbol{z}_j}(\boldsymbol{W})\right|=\left|\mathbb{E}\left[f_{\boldsymbol{z}}(\boldsymbol{W})\cdot\mathbb{1}_{\boldsymbol{Z}\neq\bar{\boldsymbol{Z}}}\right]\right|\leq\sqrt{\mathbb{E}^2f_{\boldsymbol{z}}(\boldsymbol{W})}\sqrt{\left|\mathbb{E}\mathbb{1}_{\boldsymbol{Z}\neq\bar{\boldsymbol{Z}}}\right|}
$$
$$
=\sqrt{\mathbb{E}^2f_{\boldsymbol{z}}(\boldsymbol{W})\mathbb{P}(\max_{i,j}|z_{i,j}|>B)}\leq\sqrt{R_2\mu(n,p)}. \quad (123)
$$

Let $\rho$ be the lower bound of $p$, such that $\forall p>\rho$, we have $\sqrt{R_2\mu(n,p)}<\frac{n\delta}{4}$. Hence, when $p>\rho$, we have

$$
\left|\frac{1}{p}\sum_{j=1}^{p}f_{\bar{\boldsymbol{z}}_j}(\boldsymbol{W}_l)-\mathbb{E}f_{\boldsymbol{z}_j}(\boldsymbol{W}_l)\right|\leq\left|\frac{1}{p}\sum_{j=1}^{p}f_{\bar{\boldsymbol{z}}_j}(\boldsymbol{W}_l)-\mathbb{E}f_{\bar{\boldsymbol{z}}_j}(\boldsymbol{W}_l)\right|+\frac{n\delta}{4}, \quad (124)
$$

which implies

$$
\mathbb{P}\left(\left|\frac{1}{p}\sum_{j=1}^{p}f_{\bar{\boldsymbol{z}}_j}(\boldsymbol{W})-\mathbb{E}f_{\boldsymbol{z}}(\boldsymbol{W})\right|\geq\frac{n\delta}{2}\right)\leq\mathbb{P}\left(\left|\frac{1}{p}\sum_{j=1}^{p}f_{\bar{\boldsymbol{z}}_j}(\boldsymbol{W})-\mathbb{E}f_{\bar{\boldsymbol{z}}}(\boldsymbol{W})\right|\geq\frac{n\delta}{4}\right)
$$
$$
\leq2\exp\left(-\frac{pn^2\delta^2}{32R_2+8R_1n\delta/3}\right), \quad (125)
$$

where the last inequality is achieved by Bernstein's inequality (Lemma B.1), along with assumption 3 and $\mathbb{E}f_{\bar{\boldsymbol{z}}}^2(\boldsymbol{W})\leq\mathbb{E}f_{\boldsymbol{z}}^2(\boldsymbol{W})\leq R^2$. Combine equation 125 and equation 121, we have

$$
\mathbb{P}\left(\sup_{\boldsymbol{W}\in\mathsf{O}(n;\mathbb{R})}\left|\frac{1}{p}\sum_{j=1}^{p}f_{\bar{\boldsymbol{z}}_j}(\boldsymbol{W})-\mathbb{E}f_{\boldsymbol{z}}(\boldsymbol{W})\right|\geq n\delta\right)
$$

$$
<\exp\left[n^2\ln\left(\frac{12(L_f+\bar{L}_f)}{n\delta}\right)\right]\mathbb{P}\left(\sup_{\boldsymbol{W}\in\mathbb{B}(\boldsymbol{W}_l;\epsilon)}\left|\frac{1}{p}\sum_{j=1}^{p}f_{\bar{\boldsymbol{z}}_j}(\boldsymbol{W}_l)-\mathbb{E}f_{\boldsymbol{z}}(\boldsymbol{W}_l)\right|\geq\frac{n\delta}{2}\right) \quad (126)
$$

$$
<\exp\left[-\frac{pn^2\delta^2}{32R_2+8R_1n\delta/3}+n^2\ln\left(\frac{12(L_f+\bar{L}_f)}{n\delta}\right)+\ln 2\right].
$$

**Summary.** Therefore, we conclude that

$$\mathbb{P}\left(\sup_{\boldsymbol{W}\in\mathsf{O}(n;\mathbb{R})}\frac{1}{np}\left|\sum_{j=1}^{p}\left[f_{\boldsymbol{z}_j}(\boldsymbol{W})-\mathbb{E}f_{\boldsymbol{z}_j}(\boldsymbol{W})\right]\right|\geq\delta\right)$$
$$< \exp\left[-\frac{pn^2\delta^2}{32R_2+8R_1n\delta/3}+n^2\ln\left(\frac{12(L_f+\bar{L}_f)}{n\delta}\right)+\ln 2\right]+\mu(n,p),$$

(127)

when $p > \rho$. ∎

**Lemma B.5 (Concentration Bound of the Clean Objective over $\mathsf{O}(n;\mathbb{R})$)** $\forall\theta\in(0,1]$, *if* $\boldsymbol{X}\in\mathbb{R}^{n\times p}$, $x_{i,j}\sim_{iid}BG(\theta)$, *for any* $\delta>0$, *the following inequality holds*

$$\mathbb{P}\left(\sup_{\boldsymbol{W}\in\mathsf{O}(n;\mathbb{R})}\frac{1}{np}\left|\|\boldsymbol{W}\boldsymbol{X}\|_4^4-\mathbb{E}\|\boldsymbol{W}\boldsymbol{X}\|_4^4\right|\geq\delta\right)$$
$$< \exp\left(-\frac{3p\delta^2}{c_1\theta+8n(\ln p)^4\delta}+n^2\ln\left(\frac{60np(\ln p)^4}{\delta}\right)\right)$$
$$+ \exp\left(-\frac{p\delta^2}{c_2\theta}+n^2\ln\left(\frac{60np(\ln p)^4}{\delta}\right)\right)+2np\theta\exp\left(-\frac{(\ln p)^2}{2}\right),$$

(128)

*for some constants* $c_1>10^4, c_2>3360$. *Moreover*

$$\exp\left(-\frac{3p\delta^2}{c_1\theta+8n(\ln p)^4\delta}+n^2\ln\left(\frac{60np(\ln p)^4}{\delta}\right)\right)$$
$$+ \exp\left(-\frac{p\delta^2}{c_2\theta}+n^2\ln\left(\frac{60np(\ln p)^4}{\delta}\right)\right)+2np\theta\exp\left(-\frac{(\ln p)^2}{2}\right)\leq\frac{1}{p},$$

(129)

*when* $p=\Omega(\theta n^2\ln n/\delta^2)$.

**Proof** See Lemma 2.2 in Zhai et al. (2019b), note that the sparsity condition $\theta\in(0,1)$ of the original Lemma in Zhai et al. (2019b) can be easily generalized to $\theta=1$. ∎

## C ADDITIONAL EXPERIMENTAL RESULTS

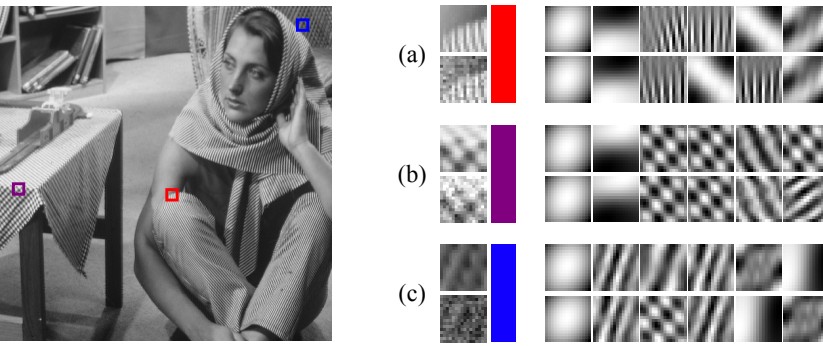

Figure 8: Representations of three $16\times16$ patches in both the clean and noisy images. Each selected patch is visualized, both with and without noise, and the 6 corresponding bases with largest absolute coefficients are shown.

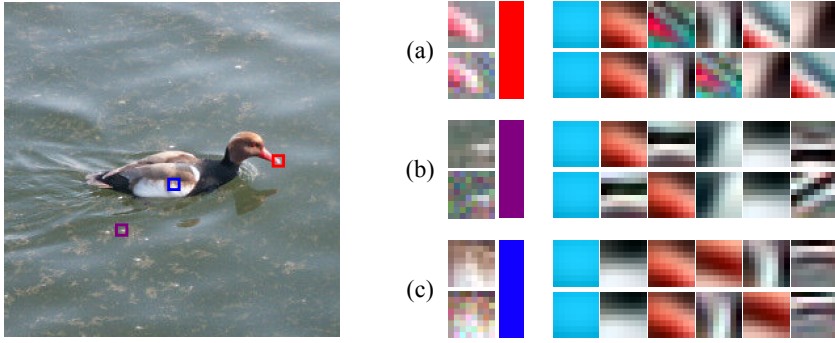

Figure 9: Representations of three $8 \times 8 \times 3$ colored patches in both the clean and noisy images. Each selected patch is visualized, both with and without noise, and the 6 corresponding bases with largest absolute coefficients are shown.

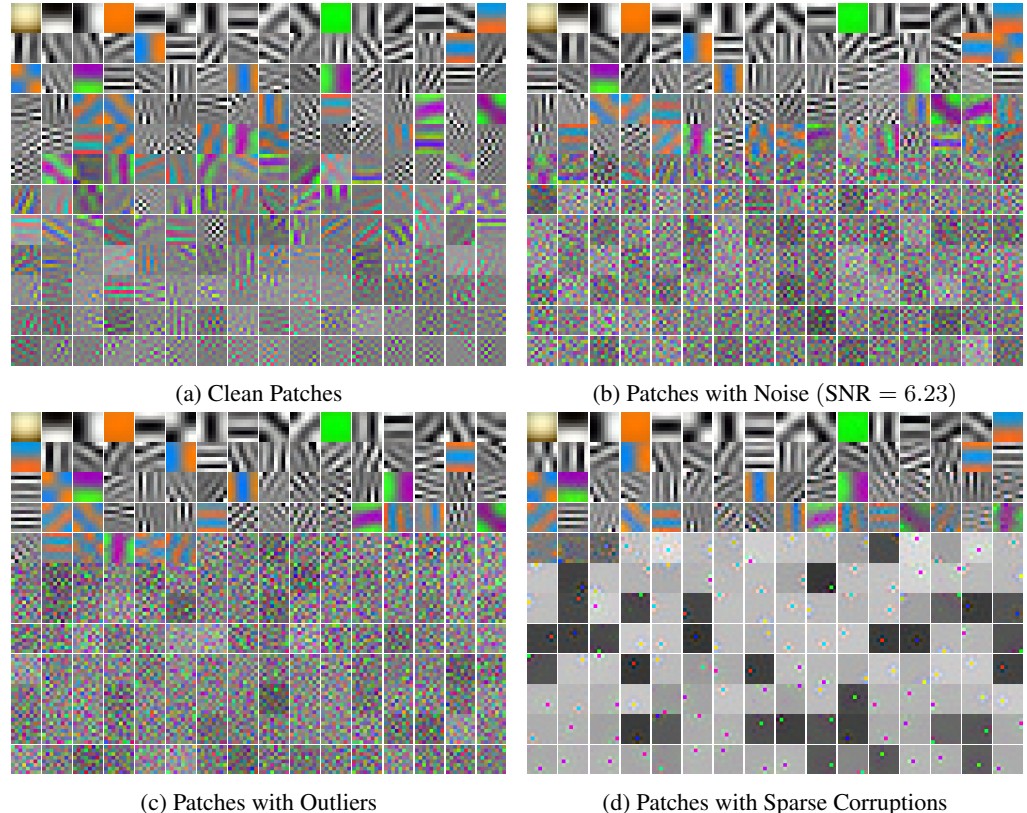

(a) Clean Patches

(b) Patches with Noise (SNR $= 6.23$)

(c) Patches with Outliers

(d) Patches with Sparse Corruptions

Figure 10: All $8 \times 8 \times 3 = 192$ bases learned from $100,000$ random $8 \times 8$ colored patches sampled from the CIFAR-10 data-set. (a) Learned Bases from clean CIFAR-10; (b) Learned Bases from CIFAR-10 with Gaussian noise, SNR $= 6.23$; (c) Learned Bases from CIFAR-10 with 20% of Gaussian outliers; (d) Learned Bases from CIFAR-10 with 50% of sparse corruptions. For all learned bases, the resulting atoms are sorted according to the $\ell^1$-norm of their coefficients in the sparse code.

