# OpenReview forum: "Understanding l4-based Dictionary Learning: Interpretation, Stability, and Robustness"
_ICLR.cc/2020/Conference — Accept (Poster)_

### Official Review · AnonReviewer1 · 2019-10-23
**Official Blind Review #1**

**Rating:** 6

**Review:**

This paper explores the recently proposed $\ell^4$-norm maximization approach for solving the sparse dictionary learning (SDL) problem. Unlike other previously proposed methods that recover the dictionary one row/column at a time, for an orthonormal dictionary, the $\ell^4$-norm maximization approach is known to recover the entire dictionary once for all.

This paper shows that $\ell^4$-norm maximization has close connections with the PCA and ICA problem. Furthermore, focusing on the MSP algorithm for solving the  $\ell^4$-norm maximization formulation, the paper highlights the connections of this fixed-point style algorithm with such algorithms for PCA and ICA. Subsequently, the paper studies the behavior of the MSP algorithm in the presence of noise, outliers, and sparse corruption. Unlike PCA, surprisingly, the MSP algorithm is shown to be robust to outliers and sparse corruption.

Overall, the paper makes a nice effort towards better understanding the relatively new $\ell^4$-norm maximization approach and its connection with other well-understood problems in the literature. Moreover, the paper takes the right step by studying the effect of non-ideal signal measurements on the underlying goal of dictionary learning. That said, the reviewer feels that, in the current form, the results in the paper are not novel enough to warrant an acceptance to ICLR. The connection of the $\ell^4$-norm maximization formulation with ICA have been previously noted in other paper, so this would hardly qualify as a novel contribution. The analysis of the MSP algorithm in the presence of noise, outlier, and sparse corruption is not comprehensive enough. It would have been nice if the authors had provided a non-asymptotic analysis of the MSP algorithm in the presence of non-ideal measurements. Also, it is not clear how interesting the outlier formulation presented in the paper is. Shouldn't one consider outliers that go beyond the Gaussian distribution, ideally arbitrary outliers?

**Experience Assessment:**

I have published one or two papers in this area.

**Review Assessment: Checking Correctness Of Derivations And Theory:**

I carefully checked the derivations and theory.

**Review Assessment: Checking Correctness Of Experiments:**

I carefully checked the experiments.

**Review Assessment: Thoroughness In Paper Reading:**

I read the paper thoroughly.

---

> ### Author Response · Authors · 2019-11-07
> **Thanks & Will extend non-asymptotic analysis**
>
> Thanks for your detailed review and for your overall positive evaluation! In what follows, we provide more detailed responses to each of your comments. We have also performed more analyses per your subsequent suggestions. And we hope you find the updated draft adequately addresses your concerns.
>
> We should have made our novelty more clear and hope to clarify the point here. We are aware that Zhai et al. [1] has already pointed out the connection between $\ell^4$-maximization and ICA. However, this connection is merely at the formulation level and is *not* a novelty we claim for this paper. Instead, our work goes beyond and establishes the connections more at the algorithmic level and in particular, our work provides a unified understanding on how such efficient power-iteration like algorithms (FastICA and MSP) can be established under the unified framework of maximizing a convex function over a compact set. We consider this unification as valuable for the community, since the general result for power methods in maximizing convex function over a compact set only appears recently Journee et al. [2] and we believe this unified view will encourage further research in this direction.
>
> Thanks for the suggestions on on which our paper can be improved with non-asymptotic concentration results. We did not include such results in the first submission due to limited space (as we prepared the paper for a 8-page version), but we already know there is no technical difficulty in reaching such results (which we have mentioned in footnote 9 of our paper.) As per your suggestion, we will provide non-asymptotic measure concentration results of the MSP algorithm in the updated version.
>
> In addition, we will also provide some extra clarifications in the regime of non-Gaussian outliers in the updated version, please stay tuned.
>
> Thanks again for your thoughtful comments, we will try our best to clarify all your concerns about our paper in the coming version.
>
> References:
> [1] Zhai, Yuexiang, Zitong Yang, Zhenyu Liao, John Wright, and Yi Ma. "Complete Dictionary Learning via $\ell^ 4$-Norm Maximization over the Orthogonal Group." arXiv preprint arXiv:1906.02435, 2019
> [2] Journée, Michel, Yurii Nesterov, Peter Richtárik, and Rodolphe Sepulchre. "Generalized power method for sparse principal component analysis." Journal of Machine Learning Research 11, no. Feb (2010): 517 - 553.

---

### Official Review · AnonReviewer3 · 2019-10-27
**Official Blind Review #3**

**Rating:** 8

**Review:**

This paper presents results on Dictionary Learning through l4 maximization. The authors base this paper heavily off of the formulation and algorithm in Zhai et. al. (2019) "Complete dictionary learning via l4-norm maximization over the orthogonal group". The paper draws connections between complete dictionary learning, PCA, and ICA by pointing out similarities between the objectives functions that are optimized as well as the algorithms used. The paper further presents results on dictionary learning in the presence of different types of noise (AWGN, sparse corruptions, outliers) and show that the l4 objective is robust to different types of noise. Finally the authors apply different types of noise to synthetic and real images and show that the dictionaries that they learn are robust to the noise applied.

Overall this paper makes significant contributions by extending the work in the paper referenced above to noisy dictionary learning settings and I would vote to accept based on these results.

The connections between Complete Dictionary Learning, PCA and ICA are interesting, but the algorithmic analogies seem superficial in my opinion. There are a lot of algorithms which follow a projected/proximal gradient descent scheme. If there are any deeper connections between the specific algorithms discussed, they should be spelled out more clearly. One point of clarification that I would like to raise is the similarity between the kurtosis and l4 objectives. This paper could be strengthened by delineating the conditions under which one would learn an ICA basis vs a Complete Dictionary. It seems to me that the only difference is in the generative model, and that maximizing the same objective under different data conditions could return an ICA basis or a Complete Dictionary.

The robustness theory and experiments on synthetic data are reasonable and demonstrate that complete dictionary learning is robust to the different noise conditions. I would like to how this technique compares to other complete dictionary learning algorithms (ER-SpUD, Complete dictionary learning over the sphere - Sun, Qu, Wright 2015) and whether the l4 objective is unique in providing this robustness. Another central claim of Zhai et. al. 2019 seems to be that l4 maximization is able to recover the entire dictionary at once, vs other algorithms that recover the dictionary one column at a time. To test this, I would like to see runtime evaluations and comparisons to other algorithms. While the claim of recovering the entire dictionary is true, it seems to me that requiring an SVD at each iteration would be very expensive. I am not completely convinced that the approach of estimating the entire dictionary would indeed be faster.

To summarize, I believe this paper would be a good addition to the literature on l4 maximization algorithms for dictionary learning. I am willing to adjust my score based on responses to the above concerns.

**Experience Assessment:**

I have published one or two papers in this area.

**Review Assessment: Checking Correctness Of Derivations And Theory:**

I carefully checked the derivations and theory.

**Review Assessment: Checking Correctness Of Experiments:**

I assessed the sensibility of the experiments.

**Review Assessment: Thoroughness In Paper Reading:**

I read the paper at least twice and used my best judgement in assessing the paper.

---

> ### Author Response · Authors · 2019-11-07
> **Thanks & Will update draft accordingly**
>
> Thank you for your detailed reading and for your positive opinion! Below, we provide a point-to-point response and hope to address your concerns.
>
> Yes, we could have made the connection point more clear. Certainly, as you pointed out, there are a lot of algorithms which follow a projected/proximal gradient descent schemes. However, there is something interesting and nontrivial here: among all projected/proximal gradient descent methods, MSP algorithms lie in the optimization regime that allows the step-size to be infinite (i.e. MSP acts like a power iteration method) and hence resulting in more efficient algorithms than the traditional gradient descent type methods. Moreover, we make the comparison between MSP, Power-iteration, and FastICA (as stated in Table 1 of the paper) to illustrate the intuition behind the efficiency of the MSP algorithm.
>
> We highly appreciate the ``ICA-basis-versus-dictionary-learning" comment and it is also very surprising for us to see how $\ell^4$-norm maximization can be derived and justified from different generative models -- ICA and Dictionary Learning. Qualitatively, we think such similarity comes from non-Gaussian property of the sparsity assumption -- as maximizing kurtosis promotes non-Gaussianity of the data which coincides with the sparse ground truth of Dictionary Learning problem. We will include a discussion on our intuition on why this occurs, because we certainly agree with you this is an intriguing phenomenon. Characterizing the exact conditions under which they coincide may be beyond the scope of the current paper and will leave the more quantitative analysis for future work.
>
> Thank you for pointing this out, a very good point that we need to demonstrate. In the updated draft, we will provide more comparisons between the $\ell^4$ formulation and the previous $\ell^1$ based methods in terms of robustness.
>
>  It is known that $\ell^1$ minimization itself is not robust to noise or outliers (hence many works in the literature on Lasso for noise measurements and error correction for $\ell^1$ minimization, see paper Wright et al. [1] and references therein.) In addition, learning the dictionary column by column is less robust than learning the entire dictionary holistically, as the error may propagate while the latter can leverage global information to denoise much more effectively.
>
> Regarding the run-time concerns, this is another good point that we did not address in the initial submission and thank you for the suggestion. We will also include run-time comparisons in the updated draft.
>
> Thank you again for your review! Per your comments, we will update our paper according to your advice, please stay tuned.
>
> We hope we have addressed all your concerns.
>
> References:
> [1] Wright, John, and Yi Ma. "Dense Error Correction Via $\ell^ 1$-Minimization." IEEE Transactions on Information Theory 56.7 (2010): 3540-3560.

---

### Decision · Program_Chairs · 2019-12-19

**Decision:**

Accept (Poster)

**Comment:**

Main content:

Blind review #3 summarizes it well:

This paper presents results on Dictionary Learning through l4 maximization. The authors base this paper heavily off of the formulation and algorithm in Zhai et. al. (2019) "Complete dictionary learning via l4-norm maximization over the orthogonal group". The paper draws connections between complete dictionary learning, PCA, and ICA by pointing out similarities between the objectives functions that are optimized as well as the algorithms used. The paper further presents results on dictionary learning in the presence of different types of noise (AWGN, sparse corruptions, outliers) and show that the l4 objective is robust to different types of noise. Finally the authors apply different types of noise to synthetic and real images and show that the dictionaries that they learn are robust to the noise applied.

--

Discussion:

Reviews agree about the interesting work, including the connections of complete dictionary learning with classic PCA and ICA (after further clarification during the rebuttal period). Additional empirical strengthening during the rebuttal period also addressed a reviewer concern.

--

Recommendation and justification:

As review #3 wrote, "Overall this paper makes significant contributions by extending the work in [Zhai et. al's (2019) "Complete dictionary learning via l4-norm maximization over the orthogonal group"] to noisy dictionary learning settings".